# ROBUST ADVERSARIAL POLICY OPTIMIZATION UNDER DYNAMICS UNCERTAINTY

## ABSTRACT

Reinforcement learning (RL) policies often fail under dynamics that differ from training, a gap not fully addressed by domain randomization or existing adversarial RL methods. Distributionally robust RL provides a formal remedy but still relies on surrogate adversaries to approximate intractable primal problems, leaving blind spots that potentially cause instability and over-conservatism. We propose a dual formulation that directly exposes the robustness–performance trade-off. At the trajectory level, a temperature parameter from the dual problem is approximated with an adversarial network, yielding efficient and stable worst-case rollouts within a divergence bound. At the model level, we employ Boltzmann reweighting over dynamics ensembles, focusing on more adverse environments to the current policy rather than uniform sampling. The two components act independently and complement each other: trajectory-level steering ensures robust rollouts, while model-level sampling provides policy-sensitive coverage of adverse dynamics. The resulting framework, robust adversarial policy optimization (RAPO) outperforms robust RL baselines, improving resilience to uncertainty and generalization to out-of-distribution dynamics while maintaining dual tractability.

## 1 INTRODUCTION

Reinforcement learning (RL) has made major progress in sequential decision-making, yet deployment in real domains is limited by dynamics mismatch Schulman et al. (2017); Haarnoja et al. (2018); Fujimoto et al. (2018). Policies trained in simulation often fail when transferred to real systems due to aleatoric and epistemic uncertainty, where small deviations can accumulate into catastrophic failures Morimoto & Doya (2005); Xu & Mannor (2010). This motivates robust RL: agents that sustain reliable performance under uncertainty Pinto et al. (2017); Rajeswaran et al. (2016). While much of RL research prioritizes maximizing nominal returns, deployment inevitably faces unseen environments, making robustness essential. In safety-critical settings such as robotics, driving, and finance, brittle policies are unacceptable Li et al. (2025); Radosavovic et al. (2024); Liu et al. (2020); Sallab et al. (2017); Gupta et al. (2025). Robustness also improves generalization, enabling adaptation to perturbations without retraining Smith et al. (2022); Ball et al. (2023). Designing robust RL policies is difficult Tessler et al. (2019); Abdullah et al. (2019). Naïve methods often yield over-conservative policies or unstable training. The challenge lies in dynamics uncertainty: optimizing only for the nominal model gives fragile policies, while optimizing for all possible models is overly pessimistic Tobin et al. (2017). The central question is: *How can we design RL algorithms robust to dynamics uncertainty, without collapsing into conservatism or instability?*

Our approach begins from the robust MDP (RMDP) formulation with dynamics uncertain within a KL neighborhood of a nominal model Wang & Zou (2021); Ho et al. (2022). The dual problem converts this infinite-dimensional problem into a tractable scalar optimization, yielding closed-form adversarial distributions, entropic variance control, and contractive operators. This corresponds to evaluating returns under an exponentially tilted trajectory law with inverse temperature $\eta$. Direct sampling from this tilted law is infeasible in simulators that only roll out the nominal kernel Pinto et al. (2017); Zhou et al. (2023). We therefore introduce an adversarial network (AdvNet) that predicts trajectory-level dual parameters. By steering rollouts toward low-return regions consistent with the KL budget, AdvNet effectively reweights trajectories to emphasize adverse outcomes, a practical realization of exponential tilting that exposes local vulnerabilities the policy might ignore. AdvNet alone cannot address global model misspecification. We therefore apply Boltzmann reweighting

over an ensemble of environments with varied physical parameters, emphasizing models where the policy underperforms while remaining close to a prior. This provides structured coverage of adverse dynamics. Together, AdvNet and Boltzmann reweighting complement one another: the former enforces robustness at the *trajectory-level*, the latter at the *model-level*. Prior approaches capture only fragments of this picture. Domain randomization (model-level) spreads training across perturbations but treats all equally, producing inefficient or cautious policies Tobin et al. (2017). Pure adversarial perturbations emphasize worst cases but lack distributional grounding and stability Pinto et al. (2017); Rigter et al. (2022); Vinitsky et al. (2020). Ensemble methods capture variability but without adversarial reweighting underestimate risk Rajeswaran et al. (2016).

Thus, our work introduces **robust adversarial policy optimization (RAPO)**, a practical realization of distributionally robust RL. RAPO combines trajectory-level exponential tilting via AdvNet with model-level Boltzmann reweighting, forming a two-layer structure that guards against local fragility and global misspecification. Finally, RAPO is practical and scalable: it amortizes trajectory tilting and ensemble biasing within a PPO-compatible loop, yielding policies that preserve in-distribution (ID) performance while substantially improving robustness to dynamics uncertainty and out-of-distribution (OOD) perturbations.

## 2 RELATED WORK

Robust MDPs model dynamics uncertainty by assuming the transition kernel lies in an ambiguity set, with policies optimized for worst-case return Morimoto & Doya (2005); Xu & Mannor (2010); Mannor et al. (2016). The power of this approach depends on how the set is defined. Early norm- or interval-bounded sets poorly captured mismatch, while distributional neighborhoods such as Wasserstein or $f$-divergence balls better align with statistical uncertainty and connect to distributionally robust optimization, yielding stronger guarantees Xu & Mannor (2010); Abdullah et al. (2019). In practice, exact evaluation requires costly nested minimax problems, creating computational and sample inefficiency Wang & Zou (2021; 2022). Many works thus approximate the objective, trading nominal performance for conservative guarantees and exposing the core tension of robust RL: rigor versus scalability Eysenbach & Levine (2021); Blanchet et al. (2023).

An alternative is adversarial RL, formulating problem as a two-player game where an auxiliary agent perturbs states, actions, or dynamics to minimize return Pinto et al. (2017); Vinitsky et al. (2020). This zero-sum view parallels adversarial training and $H_\infty$ control Morimoto & Doya (2005). Such adversaries act as disturbance generators or hard-example miners, but often yield instability, oscillatory dynamics, and narrow local robustness without distributional guarantees Tessler et al. (2019); Eysenbach & Levine (2021). Other variants perturb environment parameters, observations, or include explicit opponents in multi-agent settings Rajeswaran et al. (2016); Mankowitz et al. (2019); Zhang et al. (2020; 2021); Oikarinen et al. (2021).

Domain randomization instead samples environment parameters from broad distributions, enabling sim-to-real transfer but often producing inefficient or conservative policies Tobin et al. (2017); Peng et al. (2018); Li et al. (2025); Cai et al. (2025). Complementary research directions include verification methods with certified robustness to bounded noise Lütjens et al. (2020), adversarial input perturbations that enforce Lipschitz regularity Zhang et al. (2021), and multi-agent opponents Gleave et al. (2019); Vinitsky et al. (2020). These approaches extend robustness but target input or agent-level perturbations, remaining orthogonal to dynamics-level uncertainty.

## 3 PRELIMINARIES

**Notations:** For any finite set $\mathcal{X}$, we denote its cardinality by $|\mathcal{X}|$. The set $[n]$ represents $\{1, \ldots, n\}$. For a set $\mathcal{X}$, we write $\Delta_{\mathcal{X}}$ for the set of probability distributions over $\mathcal{X}$ (when $\mathcal{X}$ is finite, it coincides with the probability simplex), and $\mathrm{Unif}(\mathcal{X})$ for the uniform distribution on $\mathcal{X}$.

**Markov decision process:** A Markov decision process (MDP) is a tuple $(\mathcal{S}, \mathcal{A}, p, r, \mu_0, \gamma)$, where $\mathcal{S}$ is the state space, $\mathcal{A}$ the action space, $p : \mathcal{S} \times \mathcal{A} \to \Delta_{\mathcal{S}}$ the transition kernel, $r : \mathcal{S} \times \mathcal{A} \to [0, 1]$ a bounded reward function, $\mu_0 \in \Delta_{\mathcal{S}}$ the initial state distribution, and $\gamma \in (0, 1)$ the discount factor. A stochastic policy $\pi : \mathcal{S} \to \Delta_{\mathcal{A}}$ induces the expected discounted return $J(\pi, p) = \mathbb{E}_{s_0 \sim \mu_0, a_t \sim \pi, s_{t+1} \sim p}\left[\sum_{t=0}^{\infty} \gamma^t r(s_t, a_t)\right]$. The state value, state–action value, and advantage are

---

**Algorithm 1** Robust Adversarial Policy Optimization

---

**Input:** horizon $T$, ensemble size $K$, dual budget $\epsilon$, ensemble budget $\kappa$, batch size $B$, $m$ samples
**Initialize:** actor $\theta^0$, critic $\phi^0$, AdvNet $\psi^0$, prior $w_0$, weights $w \leftarrow w_0$

1: **for** $t = 0, 1, \ldots, T - 1$ **do**
2:      Collect on-policy rollout $\{(s, a, r, d)\}_{1:B}$ under nominal env
3:      $(y, \psi^{t+1}) \leftarrow \text{ADVNET-TARGETS}(\phi^t, \psi^t, w, \epsilon, m)$          // Alg. 2
4:      Update actor–critic by PPO with targets $y$ (advantages via GAE)
5:      $w \leftarrow \text{BOLTZMANN-REWEIGHTING}(w_0, \phi^{t+1}, \kappa)$          // Alg. 3
6: **end for**
7: **return** $\theta, \phi, \psi$

---

$V^\pi(s) = \mathbb{E}_{a \sim \pi(\cdot|s), s' \sim p(\cdot|s,a)}[r(s,a) + \gamma V^\pi(s')]$, $Q^\pi(s,a) = r(s,a) + \gamma \mathbb{E}_{s' \sim p(\cdot|s,a)} V^\pi(s')$, and $A^\pi(s,a) = Q^\pi(s,a) - V^\pi(s)$, respectively. The Bellman operator w.r.t. $\pi$ is $\mathcal{T}^\pi V(s) = \mathbb{E}_{a \sim \pi(\cdot|s), s' \sim p(\cdot|s,a)}[r(s,a) + \gamma V(s')]$ Sutton et al. (1998); Agarwal et al. (2019).

**Robust RL.** In many applications, the ground-truth kernel $p^\star$ is unknown or subject to perturbations. To formalize robustness, we introduce a KL-ball ambiguity set around a nominal model $\hat{p}$, which represents the transition kernel of the training environment Tamar et al. (2014); Roy et al. (2017):

$$\mathcal{P}_\epsilon(s,a) = \Big\{ p(\cdot|s,a) \in \Delta_\mathcal{S} \ : \ D_{\text{KL}}\big(p(\cdot|s,a) \,\|\, \hat{p}(\cdot|s,a)\big) \leq \epsilon \Big\}, \tag{1}$$

and $\mathcal{P}_\epsilon = \bigotimes_{s,a} \mathcal{P}_\epsilon(s,a)$ under $(s,a)$-rectangularity Mannor et al. (2016).

RMDPs then pose the following control problem: $\pi^\star \in \arg\max_\pi \inf_{p \in \mathcal{P}_\epsilon} J(\pi, p)$. Rectangularity decouples uncertainty across state–action pairs and enables dynamic programming Iyengar (2005); Nilim & El Ghaoui (2005). Given $\mathcal{P}_\epsilon$, the robust value of a policy $\pi$ is $V^\pi_{\mathcal{P}_\epsilon}(s) = \inf_{p \in \mathcal{P}_\epsilon} V^\pi_p(s)$, and the corresponding robust evaluation operator $\mathcal{T}^\pi_{\mathcal{P}_\epsilon} : \mathbb{R}^{|\mathcal{S}|} \to \mathbb{R}^{|\mathcal{S}|}$ is

$$(\mathcal{T}^\pi_{\mathcal{P}_\epsilon} V)(s) = \mathbb{E}_{a \sim \pi(\cdot|s)}\Big[r(s,a) + \gamma \inf_{p \in \mathcal{P}_\epsilon} \mathbb{E}_{s' \sim p(\cdot|s,a)} V(s')\Big]. \tag{2}$$

The robust Bellman equation $V = \mathcal{T}^\pi_{\mathcal{P}_\epsilon} V$ has the unique solution $V = V^\pi_{\mathcal{P}_\epsilon}$ under discounting and rectangularity. Under standard conditions—$(s,a)$-rectangularity, $\gamma < 1$, bounded rewards, and compact/convex uncertainty sets—both $\mathcal{T}^\pi_{\mathcal{P}_\epsilon}$ and the optimality operator $\mathcal{T}_{\mathcal{P}_\epsilon}$ are $\gamma$-contractions in $\|\cdot\|_\infty$, so $V^\pi_{\mathcal{P}_\epsilon}$ and $V^\star$ are their unique fixed points and the value iteration converges to $V^\star$. Moreover, there exists a stationary optimal policy $\pi^\star$ that achieves this robust value uniformly, i.e., $V^{\pi^\star}_{\mathcal{P}_\epsilon}(s) = \sup\{V^\pi_{\mathcal{P}_\epsilon}(s) : \pi \text{ history-dependent}\}$ for all $s \in \mathcal{S}$ Iyengar (2005); Nilim & El Ghaoui (2005). Thus, it suffices to optimize over stationary policies. For any stationary $\pi$, there exists a stationary worst-case kernel $p^{\text{wc}}_\pi \in \mathcal{P}_\epsilon$ such that $V^\pi_{\mathcal{P}_\epsilon}(s) = V^\pi_{p^{\text{wc}}_\pi}(s)$ for all $s$. We define the robust state–action value and advantage as $Q^\pi_{\mathcal{P}_\epsilon}(s,a) := Q^\pi_{p^{\text{wc}}_\pi}(s,a)$ and $A^\pi_{\mathcal{P}_\epsilon}(s,a) := A^\pi_{p^{\text{wc}}_\pi}(s,a)$, omitting the subscript $\mathcal{P}_\epsilon$ when clear from context. Deterministic stationary optimality follows since the maximization is linear in the action distribution, so the optimum lies at an extreme point of the simplex.

**Occupancy and performance difference.** For a stationary policy $\pi$, the discounted occupancy measure is defined as $d^\pi(s) = (1 - \gamma) \sum_{t \geq 0} \gamma^t \Pr_\pi(s_t = s)$, which specifies how frequently each state is visited under $\pi$ (in continuous state spaces, the definition extends by treating $d^\pi$ as a measure). The performance difference lemma (PDL) states that for any $\pi, \pi'$, the return difference satisfies $J(\pi') - J(\pi) = \frac{1}{1-\gamma} \mathbb{E}_{s \sim d^{\pi'}, a \sim \pi'(\cdot|s)}[A^\pi(s,a)]$, i.e., improvement is the advantage of $\pi$ averaged under the new policy's occupancy Kakade & Langford (2002). In the robust setting, let $p^{\text{wc}}_\pi \in \mathcal{P}_\epsilon$ be a stationary worst-case kernel so that $V^\pi_{\mathcal{P}_\epsilon} = V^\pi_{p^{\text{wc}}_\pi}$ (under rectangularity and contraction). Define the robust discounted occupancy of $\pi'$ by $d^{\pi'}_{\mathcal{P}_\epsilon}(s) = (1 - \gamma) \sum_{t \geq 0} \gamma^t \Pr_{(p^{\text{wc}}_\pi, \pi')}(s_t = s)$. Write $Q^\pi_{\mathcal{P}_\epsilon}(s,a) := Q^\pi_{p^{\text{wc}}_\pi}(s,a)$ and $A^\pi_{\mathcal{P}_\epsilon}(s,a) := A^\pi_{p^{\text{wc}}_\pi}(s,a)$. The robust performance difference lemma (RPDL) then gives $J(\pi', p^{\text{wc}}_{\pi'}) - J(\pi, p^{\text{wc}}_\pi) = \frac{1}{1-\gamma} \mathbb{E}_{s \sim d^{\pi'}_{\mathcal{P}_\epsilon}, a \sim \pi'(\cdot|s)}[A^\pi_{\mathcal{P}_\epsilon}(s,a)]$, showing that robust improvement is exactly the expected robust advantage under the new policy's robust occupancy. See App. G for proofs and further details.

## 4 ROBUST ADVERSARIAL POLICY OPTIMIZATION (RAPO)

### 4.1 ADVERSARIAL NETWORK FOR DUAL TEMPERATURE ESTIMATION

**Dual representation and the $\eta$ trade-off.** The robust formulation begins from the hard-constraint primal in 2 and, via convex duality, admits the equivalent dual form

$$\inf_{p \in \mathcal{P}_\epsilon} \mathbb{E}_p[V] = \sup_{\eta \geq 0} \left\{ -\tfrac{1}{\eta} \log \mathbb{E}_{\hat{p}}[e^{-\eta V}] - \tfrac{\epsilon}{\eta} \right\}, \tag{3}$$

where $\hat{p}$ is the nominal kernel and $\eta$ is the dual temperature that exponentially tilts returns. The offset term $-\epsilon/\eta$ reflects the KL budget and ensures robustness guarantees. The parameter $\eta$ controls the trade-off: larger $\eta$ sharpens focus on adverse outcomes, but also drives divergence from the nominal prior. In practice, $\hat{p}$ corresponds to the training environment's transition kernel. To capture parametric variability, we instantiate an ensemble of perturbed kernels around $\hat{p}$, and Boltzmann reweighting adjusts their mixture weights under a KL budget. This extends the nominal model into an ensemble-based prior while retaining the dual interpretation. In particular, the dual form above computes the inner minimization in 2. See App. A for the full derivation.

**Why dual instead of primal?** A natural question arises as to why the dual formulation is employed instead of directly optimizing the primal objective $\inf_{p: D_{\mathrm{KL}}(p\|\hat{p}) \leq \epsilon} \mathbb{E}_p[V]$. Primal-based approaches, including adversarial training methods such as RARL Pinto et al. (2017), adversarial policies Gleave et al. (2019), or sampling-driven methods like RNAC Zhou et al. (2023), attempt to approximate robustness by explicitly searching over a family of perturbed environments or by introducing an adversarial agent. In practice, this reduces to evaluating a finite set of candidate dynamics models. However, such a construction cannot guarantee coverage of the entire ambiguity set: the worst-case distribution may lie in between sampled models, leaving blind spots, or else lead to instability and over-conservatism due to reliance on extreme samples. By contrast, the dual form 3 collapses the infinite-dimensional optimization over distributions into a single scalar parameter $\eta$. This converts the robust objective into a tractable *risk-sensitive expectation* under the nominal model, where the adversarial distribution is implicitly realized through exponential tilting rather than explicit sampling. Ensemble models in RAPO are only used to approximate the nominal distribution $\hat{p}$ and to provide epistemic variability, while robustness is guaranteed beyond the finite set of sampled environments. Moreover, the scalar $\eta$ functions as a principled robustness–performance knob, enabling an efficient and interpretable trade-off that is absent in primal formulations.

**AdvNet: amortized approximation of $\eta^\star$.** Direct sampling from distribution $p_\eta(s' \mid s, a) = e^{-\eta V(s')} / \sum_j e^{-\eta V(s'_j)}$ is infeasible in deterministic simulators, and solving for $\eta^\star$ by root finding each step is computationally prohibitive. To overcome this, we introduce an adversarial network (AdvNet) that amortizes the search for $\eta^\star$ by predicting it directly from rollouts (Alg. 2). AdvNet is a feedforward network that takes as input the current state $s_t$ and optionally the action $a_t$, and outputs a scalar $\eta_t = \mathrm{AdvNet}_\psi(s_t, a_t) \geq 0$, with a softplus activation ensuring nonnegativity. Further discussion on state-action dependency of $\eta$ is in App. C. Given $m$ samples $s'_i \sim \hat{p}(\cdot \mid s, a)$ with values $V(s'_i)$, we define $\widehat{Z}_\eta = \frac{1}{m} \sum_{i=1}^m e^{-\eta V(s'_i)}$ and $v_{\mathrm{rob}}(V, \eta) = -\frac{1}{\eta} \log \widehat{Z}_\eta - \frac{\epsilon}{\eta}$. The robust Bellman target is $y_t = r_t + \gamma v_{\mathrm{rob}}(V, \eta_t)$. To enforce the KL bound, we compute $\widehat{D}_{\mathrm{KL}}(\eta) = \sum_{i=1}^m \hat{p}_i(\eta) \log(m \hat{p}_i(\eta))$ with $\hat{p}_i(\eta) = e^{-\eta V(s'_i)} / \sum_j e^{-\eta V(s'_j)}$, and penalize deviations from $\epsilon$. The loss is $\mathcal{L}_{\mathrm{AdvNet}}(\psi) = \mathbb{E}[y_t] + \lambda_{\mathrm{KL}} \mathbb{E}[(\widehat{D}_{\mathrm{KL}}(\eta_t) - \epsilon)^2_+]$, where $\lambda_{\mathrm{KL}}$ is a penalty coefficient.

**Stability and approximation guarantees.** AdvNet does not generate explicit perturbations but predicts $\eta_t$ that approximates the optimal dual temperature $\eta^\star$. This amortizes the inner maximization, avoiding costly root finding and variance, while adapting $\eta$ to the local state–action context. Importantly, the robust Bellman operator remains a $\gamma$-contraction under the supremum norm regardless of whether $\eta$ is exact or approximate. If $\hat{\eta}$ is the AdvNet output with $|\hat{\eta} - \eta^\star| \leq \delta$, then the induced value gap satisfies $\left\| \mathcal{T}^{\hat{\eta}} V - \mathcal{T}^{\eta^\star} V \right\|_\infty \leq C \delta^2$, for a constant $C$ depending on the boundedness of returns. Thus, convergence to a unique fixed point is preserved even under imperfect predictions, and approximation errors in $\eta$ only enter quadratically into the robust value update. Boltzmann reweighting, introduced next, then leverages these $\eta_t$ values to emphasize globally harmful dynamics across the ensemble. See App. C for the full derivation and proof.

---

**Algorithm 2** AdvNet-Targets (Robust one-step targets)

---

**Input:** critic $\phi$, AdvNet $\psi$, mixture $w$, KL budget $\epsilon$, samples $m$

1: Sample $m$ next-states from ensemble using $w$; evaluate critic $V_\phi$ to get $V_n \in \mathbb{R}^{B \times m}$
2: Predict temperatures $\eta_t \leftarrow g_\psi(s, a \text{ or } 0)$; project to budget $\eta^\star \leftarrow \text{proj}(V_n, \eta_t, \epsilon)$ (bisection)
3: Straight-through $\tilde{\eta} \leftarrow \eta^\star - \text{stopgrad}(\eta_t) + \eta_t$
4: Robust dual expectation $v_{\text{rob}} \leftarrow -\frac{1}{\tilde{\eta}} \log \frac{1}{m} \sum_{j=1}^{m} e^{-\tilde{\eta} V_n^{(j)}} - \frac{\epsilon}{\tilde{\eta}}$
5: Targets $y \leftarrow r + \gamma(1 - d) v_{\text{rob}}$
6: Update $\psi$ by a few Adam steps on $\mathcal{L}_{\text{AdvNet}} = \mathbb{E}[y] + \lambda_{\text{KL}} \mathbb{E}[(\widehat{D}_{\text{KL}}(\tilde{\eta}) - \epsilon)_+^2]$
7: **return** $y, \psi$

---

**Hard-minimum baseline.** A naïve approximation would be to evaluate robustness by taking the hard minimum over sampled next-state values, i.e., $\min_i V(s_i')$. However, this corresponds to the degenerate distribution $p = \delta_{s'_{\min}}$, which incurs $D_{\text{KL}}(\delta_{s'_{\min}} \| \hat{p}) = \log(1/\hat{p}(s'_{\min}))$, equaling $\log m$ under a uniform empirical distribution with $m$ samples. Unless $\epsilon \geq \log m$, the KL constraint is violated; in continuous spaces, the KL even diverges. Thus, the hard minimum is infeasible under finite KL budgets, and the correct robust solution is the *soft minimum* induced by exponential tilting.

## 4.2 BOLTZMANN REWEIGHTING OF ENVIRONMENT DISTRIBUTION

**Motivation.** Naïve sampling would draw $m$ rollouts uniformly from $K$ ensemble environments. This provides an unbiased estimator of the nominal expectation but fails to emphasize rare yet harmful dynamics. Among the ensemble, there will inevitably exist configurations where the policy is particularly vulnerable (i.e., with low value estimates). Sampling from such adverse models is more informative than uniform sampling, since it exposes the policy to weaknesses that dominate the robust objective. Formally, solving $\inf_{p \in \mathcal{P}} \mathbb{E}_p[V]$ requires Monte Carlo evaluation of expectations under $p$, which the dual formulation rewrites as an expectation under the nominal kernel $\hat{p}$. Boltzmann reweighting can therefore be interpreted as tilting $\hat{p}$ away from uniformity and toward adversarial directions, while remaining regularized by a KL budget. We specifically adopt a KL constraint since it yields closed-form Boltzmann weights and a tractable one-dimensional dual.

**Global variability and robust objective.** While AdvNet provides trajectory-level robustness by estimating dual temperatures for exponential tilting, robust evaluation also requires accounting for global variability across dynamics parameters. Directly sampling from a full probabilistic family is infeasible, so we approximate dynamics uncertainty through a finite ensemble $\{p_{\omega_k}\}_{k=1}^{K}$ with prior weights $\rho \in \Delta_{[K]}$. The goal is to emphasize harmful dynamics without discarding proximity to the prior, which motivates a Boltzmann reweighting scheme under a KL budget. We constrain adversarial weights $w \in \Delta_{[K]}$ by $\mathcal{W}_\rho(\kappa) = \{w \in \Delta_{[K]} : D_{\text{KL}}(w \| \rho) \leq \kappa\}$, and define the robust objective $J_{\text{rob}}(\pi_\theta) = \min_{w \in \mathcal{W}_\rho(\kappa)} \mathbb{E}_{\omega \sim w}[J(\pi_\theta, p_\omega)]$, where $J(\pi_\theta, p_\omega)$ is the expected discounted return of $\pi_\theta$ in dynamics $p_\omega$.

**Vulnerability scoring and Boltzmann solution.** For each model $k$, we compute a vulnerability score $h_k = \mathbb{E}_{(s,a) \sim \mathcal{D}}[V_{p_{\omega_k}}^{\pi_\theta}(s')]$ with $s' \sim p_{\omega_k}(\cdot \mid s, a)$, where $\mathcal{D}$ is the empirical rollout distribution under the current policy. Alternatives such as advantage- or cost-based surrogates are also possible. The adversary then solves $\min_{w \in \Delta_{[K]}} \sum_{k=1}^{K} w(k) h_k$ subject to $D_{\text{KL}}(w \| \rho) \leq \kappa$, whose dual yields the Boltzmann form

$$w_\beta^\star(k) = \frac{\rho(k) \exp(-\beta h_k)}{\sum_{k=1}^{K} \rho(k) \exp(-\beta h_k)}, \qquad \beta \geq 0, \tag{4}$$

with $\beta$ adjusted so $D_{\text{KL}}(w_\beta^\star \| \rho) = \kappa$. As $\beta \to 0$, $w_\beta^\star \to \rho$; as $\beta \to \infty$, $w_\beta^\star$ concentrates on the most adverse models. Numerical solutions for $\beta$ use one-dimensional search exploiting the monotonicity of the KL term. See App. B for the full derivation.

**Role of ensembles and practical considerations.** Unlike model-based planning methods that train probabilistic dynamics ensembles to capture both epistemic and aleatoric uncertainty, RAPO employs ensembles of deterministic simulators with varied dynamics parameters Chua et al. (2018); Yu et al. (2020). This construction does not aim to model intrinsic stochasticity but instead serves

---

**Algorithm 3** Boltzmann Reweighting

---

**Input:** prior $w_0 \in \Delta_{[K]}$, critic $V_\phi$, KL budget $\kappa$

1: Subsample $(s, a)$; roll out all $K$ models to get $s'_k$; score $H_k \leftarrow \frac{1}{S} \sum V_\phi(s'_k)$
2: Define $w(\beta) \propto w_0 \odot \exp(-\beta H)$
3: Find $\beta^\star \geq 0$ by bisection s.t. $D_{\mathrm{KL}}(w(\beta^\star) \| w_0) \approx \kappa$
4: **return** $w \leftarrow w(\beta^\star)$

---

as a Monte Carlo device: it approximates the nominal kernel $\hat{p}$ by aggregating deterministic transitions from multiple parameter settings. As such, RAPO primarily reflects epistemic-style variability across plausible dynamics configurations, while aleatoric uncertainty remains outside our scope. Boltzmann reweighting is therefore a principled mechanism that interpolates smoothly between the prior and worst-case emphasis while avoiding collapse. In practice, we stabilize $h_k$ with moving averages and variance reduction, and optionally regularize rapid changes in $w$. Importantly, $\eta$ here governs model-level weighting and is distinct from the trajectory-level $\eta$ predicted by AdvNet. Together, AdvNet captures local fragilities within rollouts, while Boltzmann reweighting emphasizes globally adverse dynamics across the ensemble, yielding complementary robustness. Finally, although RAPO operates with a finite ensemble $\{p_{\omega_k}\}_{k=1}^K$, App. F establishes that as $K \to \infty$ the dual objective, optimal weights, and induced policies converge to those of the continuous family at the Monte Carlo rate $O(K^{-1/2})$, providing theoretical justification for the finite approximation.

### 4.3 Unified RAPO Algorithm

RAPO unifies trajectory-level robustness from AdvNet with model-level robustness from Boltzmann reweighting into a scalable min–max policy optimization framework. At each iteration, it solves

$$\max_\theta \min_{w, \psi} \mathbb{E}_{\omega \sim w} \left[ \mathbb{E}_{\tilde{\tau} \sim (\pi_\theta, p_\omega, \psi)} \left[ \sum_{t=0}^\infty \gamma^t \, r(\tilde{s}_t, \tilde{a}_t) \right] \right], \tag{5}$$

where $\psi$ parameterizes the trajectory-level adversary (AdvNet) and $w$ is the model mixture constrained by a KL budget $\kappa$. The inner minimization couples trajectory-level exponential tilting with model-level Boltzmann reweighting, while the outer maximization updates the policy. Alg. 1 instantiates this framework, with additional stability enhancements (e.g., budget scheduling, entropy regularization) described in App. H. The joint KL constraint naturally decomposes into two entropic risks: a model-level temperature $\beta$ governing ensemble reweighting under budget $\kappa$, and a trajectory-level temperature $\eta$ controlling return tilting under budget $\epsilon$. The total robustness budget satisfies $\kappa + \epsilon \leq B$, yielding a factorized adversary with complementary roles for Boltzmann reweighting and AdvNet. This two–temperature view provides the theoretical foundation of RAPO. See App. E for the full derivation.

### 4.4 Connection to the Robust Performance Difference Lemma

RAPO builds directly on the RPDL perspective: robust policy improvement depends on expected robust advantages under the new policy's robust occupancy. In practice, however, the worst-case kernel $p_\pi^{\mathrm{wc}}$ that induces this occupancy is not directly accessible. RAPO therefore constructs a tractable surrogate distribution through a two-layer approximation: (i) *model-level Boltzmann reweighting*, which forms a KL-constrained mixture $\bar{p}_{w_\beta^\star} = \sum_k w_{\beta,k}^\star p_{\omega_k}$ (4) over a finite ensemble of dynamics, thereby emphasizing adverse models while remaining close to a prior and (ii) *trajectory-level exponential tilting*, where AdvNet predicts $\eta(s, a)$ to bias next-state sampling under $\bar{p}_{w_\beta^\star}$ toward lower-value outcomes, yielding $p_{\beta,\eta}^{\pi'}(s'|s, a) = \frac{\bar{p}_{w_\beta^\star}(s'|s,a) \exp(-\eta(s,a) V^{\pi'}(s'))}{\int \bar{p}_{w_\beta^\star}(\tilde{s}|s,a) \exp(-\eta(s,a) V^{\pi'}(\tilde{s})) \, d\tilde{s}}$. The resulting surrogate occupancy $\tilde{d}_{\beta,\eta}^{\pi'}(s) = (1 - \gamma) \sum_{t=0}^\infty \gamma^t \Pr_{(p_{\beta,\eta}^{\pi'}, \pi')}(s_t = s)$ serves as a practical replacement for $d_{\mathcal{P}_\epsilon}^{\pi'}$. Policy updates in RAPO can therefore be interpreted as optimizing robust advantages under $\tilde{d}_{\beta,\eta}^{\pi'}$, with model-level and trajectory-level robustness acting independently yet complementarily. Importantly, the discrepancy between the true and surrogate occupancies admits a clean decomposition: $|\mathbb{E}_{d_{\mathcal{P}_\epsilon}^{\pi'}}[f] - \mathbb{E}_{\tilde{d}_{\beta,\eta}^{\pi'}}[f]| \lesssim c_1 \sqrt{\kappa} + c_2 \sqrt{\epsilon} + c_3 K^{-1/2} + c_4 N^{-1/2} + c_5 \delta^2,$

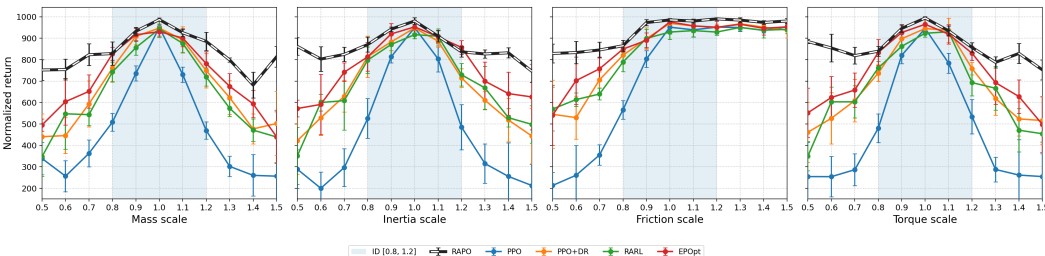

Figure 1: `Walker2d` robustness curves across mass, inertia, friction, and torque scaling. RAPO matches PPO in-distribution while significantly outperforming all baselines under OOD regions.

where $\|f\|_\infty < \infty$ and the terms correspond respectively to model-budget, trajectory-budget, finite-ensemble, finite-sample, and dual-temperature approximation errors (see App. E for derivation). Setting $f(s,a) = A^\pi_{\mathcal{P}_\epsilon}(s,a)$ recovers the RPDL, so that RAPO's surrogate occupancy provides a principled approximation for evaluating robust policy improvement. Also, the trajectory KL budget $\epsilon$ serves as a principled robustness–performance knob. From App. G, the nominal–robust value gap satisfies $|V^\pi_{\hat{p}} - V^\pi_{\mathcal{P}_\epsilon}| = O(\sqrt{\epsilon})$, showing that enlarging $\epsilon$ strengthens protection against adversarial transitions while degrading nominal performance only at a controlled $\sqrt{\epsilon}$ rate.

## 5 EXPERIMENTS

We evaluate whether RAPO improves robustness to dynamics variations while preserving ID performance comparable to standard baselines. Policies are trained under nominal dynamics but tested under perturbed parameters, so we report results in both ID (within training support) and OOD. Controlled sweeps measure robustness and performance degradation under parameter shifts. `Walker2d` is our main testbed Freeman et al. (2021). Its locomotion is sensitive to dynamics changes, making it a representative benchmark. We vary four parameter families independently: mass, inertia, friction, and torque scaling, each swept from $0.5$ to $1.5$ in steps of $0.1$, with $1.0$ as nominal. The ID region is $[0.8, 1.2]$, and values outside are OOD. Each setting is run with five seeds, reporting mean episode return with 95% confidence intervals. We also evaluate on a quadrotor payload tracking task, demonstrating RAPO's ability to handle coupled dynamics beyond standard locomotion benchmarks. Additional details and further experiments are in App. D.

### 5.1 ROBUSTNESS TO DYNAMICS UNCERTAINTY

Fig. 1 compares RAPO with PPO, PPO+DR, RARL, and EPOpt across mass, inertia, friction, and torque scaling. RAPO shows the most stable returns. We include PPO as a strong nominal baseline, PPO+DR to represent domain randomization, RARL for two-player adversarial training, and EPOpt as a risk-averse (worst-quantile/CVaR-style) objective. Together these cover nominal, randomization, adversarial, and distributionally robust strategies under matched rollout budgets and network sizes. In the ID region it closely tracks PPO with minor variation. RAPO requires more computation per step due to ensemble rollouts and Boltzmann reweighting, so it is not strictly more sample-efficient than PPO. In OOD regimes, PPO collapses once parameters leave training support, PPO+DR slows but cannot prevent degradation, RARL is unstable, and EPOpt is conservative. RAPO sustains high returns at extreme scales with only modest decline, achieving a favorable robustness–performance balance. Boltzmann reweighting emphasizes difficult regimes, sometimes producing non-monotonic effects (e.g., inertia scale $0.5$ outperforming $0.6$). Even within the ID prior, returns drop near parameter extremes, showing that uniform randomization does not yield uniform difficulty. Occasionally RAPO even outperforms PPO at the nominal scale, reflecting variance reduction from exponential tilting that stabilizes updates.

### 5.2 ROLE OF ADVNET AND BOLTZMANN REWEIGHTING

We study RAPO ablations on `Walker2d` (Fig. 2). All variants use identical budgets and differ only by enabling AdvNet and/or Boltzmann reweighting. Full RAPO achieves the highest or tied returns in ID and remains stable OOD. Within $[0.9, 1.1]$ it is close to the no-reweighting variant, but beyond

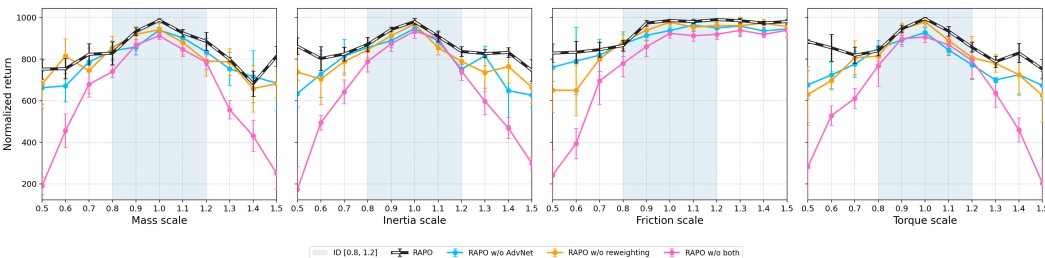

Figure 2: `Walker2d` ablation. We compare RAPO, w/o AdvNet, w/o Boltzmann reweighting ($w = \rho$), and w/o both. ID band shaded. Dropping either lowers robustness; dropping both gives the steepest OOD drop. Friction plot is flat for scales $\geq 1.0$ due to no difference in frictional force.

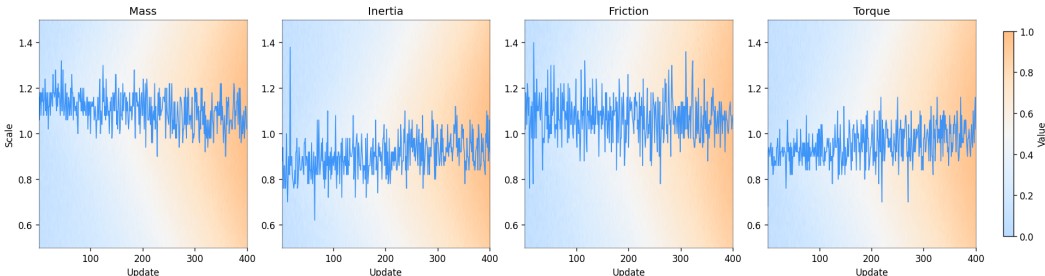

Figure 3: Heatmaps for `Walker2d`. Each panel covers only one parameter scale $\alpha$ at each with others fixed at 1.0, showing $V_u(\alpha)$ over updates $u$. RAPO lifts the value floor, while without Boltzmann reweighting ($w = Unif([K])$) leaves OOD tails. See App. D.

this band RAPO stays flat while ablations degrade. Removing both causes the steepest OOD drop, showing that uniform sampling without trajectory stress yields brittle tails. Boltzmann reweighting emphasizes difficult regimes, producing occasional non-monotonic but helpful effects (e.g., inertia 0.5 outperforming 0.6). For friction, scales $\geq 1.0$ show flat curves. *RAPO without AdvNet* removes the trajectory adversary and sets the adversarial loss to zero, instead computing $\eta^\star$ by root finding on the convex dual so that the trajectory-level KL budget $\epsilon$ is satisfied. For computational feasibility, this root finding is run for only a finite number of iterations, which makes it less effective than amortized prediction via AdvNet and generally yields weaker robustness. *RAPO without Boltzmann reweighting* fixes $w = \rho$, so environments are sampled uniformly from the entire prior distribution rather than focusing on adverse regimes, while AdvNet still perturbs trajectories. *RAPO without both* fixes $w = \rho$ and disables AdvNet, reducing training to PPO on the prior mixture with uniform sampling. All other hyperparameters and rollout settings are identical. To further illustrate robustness, Fig. 3 shows heatmaps of value estimates over training. For each update $u$ and parameter type, we sweep a dense grid of scales $\alpha \in [0.5, 1.5]$ while holding others at 1.0 and compute $V_u(\alpha)$ from one-step critic rollouts. Although training uses a finite $K$-mixture, this diagnostic is continuous. Brighter colors indicate higher value and the blue trace marks $\arg\max_\alpha V_u(\alpha)$. Under RAPO, the value floor progressively rises across the full range: early updates show dark bands at extremes (hard regimes), which flatten and brighten over time. The result shows that RAPO actively covers entire parameter space, not leaving blind spots. See App. D for further experiment results.

## 5.3 QUADROTOR PAYLOAD TRACKING EXPERIMENTS

We evaluate RAPO on harder task: flexible cable-suspended payload trajectory tracking under random force and torque disturbances. Unlike quadrotor-only tracking, payload motion adds stronger coupling and hybrid effects from the suspended mass and cable, making robustness harder. Unlike locomotion tasks, the control objective is the payload position itself, which makes trajectory visualization informative. In both `Walker2d` and quadrotor-payload system, evaluation fixes dynamics parameters per episode (e.g., mass scale or payload mass/length), so these tests assess domain-shift generalization rather than strict rectangularity. The quadrotor environment also includes per-step random force/torque disturbances, which better approximate the rectangular RMDP setting. Thus the payload system tests robustness under both episode-level shifts and step-level disturbances. A Lissajous trajectory is used for stability tests under nominal (ID) and perturbed (OOD) dynamics.

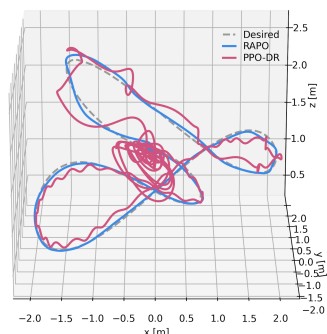 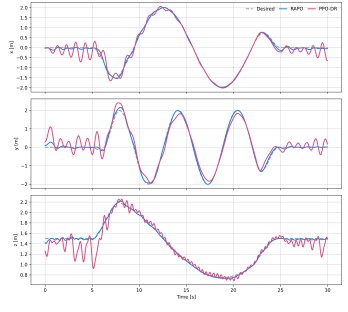

| Alg | ID | OOD | Crash [%] |
|---|---|---|---|
| PPO | 7.5 | 18.2 | 40 |
| PPO-DR | 7.8 | 14.6 | 12 |
| EPOpt | 9.1 | 13.9 | 16 |
| RARL | 8.4 | 15.2 | 25 |
| **RAPO** | 7.6 | **9.4** | **0** |

Figure 4: 3D trajectory. RAPO stays close while PPO-DR drifts in the OOD environment.

Figure 5: $x, y, z$ vs. time. RAPO reduces oscillations in the OOD environment.

Table 1: Payload tracking in the OOD environment. RMSE [cm] $\pm$ 95% CI, reported only for successful episodes. Crashes are evaluated separately from RMSE, based on 20 episodes. RAPO achieves zero crash.

For ID, payload mass is sampled from $[0, 0.25]$ kg and cable length from $[0, 0.5]$ m. For OOD, mass is $0.5$ kg and cable length $1.0$ m, producing clear shifts in inertia and tension. Also, force and torque magnitude ranges are twice greater than ID. The environment includes sensor noise, random force/torque disturbances, aerodynamic downwash, propeller time constants ($\tau_{\text{up}}, \tau_{\text{down}}$), and small initial perturbations. Fig. 4 shows the 3D trajectory: RAPO stays close to the reference, while PPO-DR drifts and oscillates, especially near the origin during hovering. PPO-DR also crashes mid-trajectory, consistent with Tab. 1. Fig. 5 shows $x, y, z$ over time: RAPO reduces oscillations and steady-state error. Performance is measured by root-mean-square error (RMSE) between desired and actual payload trajectories. Results average 10 trials with 95% confidence intervals. RAPO achieves the lowest OOD RMSE, showing that combining trajectory- and model-level adversarialization stabilizes payload dynamics. Note that observation noise differs from RMDP uncertainty: the next state $s_{t+1}$ is deterministic, while the agent observes $o_{t+1} = s_{t+1} + \epsilon$. This is closer to input perturbation or POMDP formulations than to rectangular RMDPs, which assume uncertainty in the transition kernel. Domain randomization samples dynamics per episode, fixing the kernel for the whole trajectory. This is weaker than step-wise rectangular RMDPs and can be viewed as a subset or relaxed variant of the rectangular assumption.

## 6 CONCLUSION AND FUTURE WORK

We introduced **RAPO**, a framework that realizes distributionally robust RL by combining trajectory-level exponential tilting and model-level Boltzmann reweighting within a KL-robust dual formulation. This addresses the training–deployment mismatch where RL policies degrade under unseen dynamics: unlike domain randomization or adversarial RL, distributional robustness formally accounts for such uncertainty, while prior surrogate-based primal methods suffer blind spots, instability, and over-conservatism due to incomplete coverage of the ambiguity set. RAPO builds on contraction of the robust Bellman operator, stationary optimality, and a robust performance-difference lemma, with the nominal–robust value gap scaling as $O(\sqrt{\epsilon})$, thereby interpreting $\epsilon$ as a robustness–performance knob. Practically, RAPO amortizes dual-temperature search via AdvNet to steer worst-case roll-outs efficiently and stably within divergence bounds, while Boltzmann reweighting over ensembles turns naive uniform sampling into policy-sensitive sampling, emphasizing dynamics that most expose policy vulnerabilities. These trajectory- and model-level components operate independently yet complement one another, yielding a surrogate robust occupancy distribution with controllable errors and jointly strengthening robustness. Empirically, RAPO preserves ID performance while improving OOD robustness in dynamics sweeps and quadrotor payload tracking under disturbances, reducing failure and RMSE over PPO, PPO-DR, EPOpt, and RARL. While RAPO entails additional ensemble and dual costs, approximation errors in $\eta$ enter quadratically, and total error decomposes into $\sqrt{\kappa}$ (model), $\sqrt{\epsilon}$ (trajectory), $K^{-1/2}$ (ensemble), $N^{-1/2}$ (sampling), and $\delta^2$ (dual approximation). Overall, RAPO closes the theory–practice gap by maintaining tractability while delivering robustness to uncertainty and generalization to OOD environments.

Potential future work include pre-training ensembles and priors from real-world data through safe identification, thereby mitigating the simulation-to-reality gap, and integrating RAPO with model-based planning or Dyna-style rollouts to reduce interaction cost while retaining robustness.

## ACKNOWLEDGEMENTS

We used a large language model (LLM) to assist with polishing the writing. All ideas and contributions are the authors' own. This use complies with the ICLR policy on LLM usage, and the authors take full responsibility for the content in accordance with academic integrity.

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

# A  DUAL FORMULATION OF KL-ROBUST EXPECTATION

## A.1  SETUP, PRIMAL, AND DUAL

Fix a state–action pair $(s, a)$ and a nominal transition kernel $\hat{p}(\cdot \mid s, a)$. For a bounded $V : \mathcal{S} \to \mathbb{R}$, the stage-wise robust expectation under a KL-ball defined as 1 is $\inf_{p \in \mathcal{P}_\epsilon} \mathbb{E}_{s' \sim p}[V(s')]$. Equivalently, $\min_{p \in \Delta_\mathcal{S}} \sum_x p(x) V(x)$ s.t. $\sum_x p(x) \log \frac{p(x)}{\hat{p}(x)} \leq \epsilon$, which is a convex program with strong duality by Slater's condition. Introducing multipliers $\eta \geq 0$ and $\mu \in \mathbb{R}$, the Lagrangian is $\mathcal{L}(p, \eta, \mu) = \sum_x p(x) V(x) + \eta \left( \sum_x p(x) \log \frac{p(x)}{\hat{p}(x)} - \epsilon \right) + \mu \left( \sum_x p(x) - 1 \right)$. Stationarity in $p(x)$ gives $V(x) + \eta \left( 1 + \log \frac{p(x)}{\hat{p}(x)} \right) + \mu = 0$, hence $p(x) \propto \hat{p}(x) e^{-\frac{1}{\eta} V(x)}$ and $p_\eta(x) = \frac{\hat{p}(x) e^{-\eta V(x)}}{Z(\eta)}$, $Z(\eta) = \mathbb{E}_{\hat{p}}[e^{-\eta V}]$. Substitution yields the dual $g(\eta) = -\frac{1}{\eta} \log Z(\eta) - \frac{\epsilon}{\eta}$ for $\eta \geq 0$, so the dual problem is $\max_{\eta \geq 0} -\frac{1}{\eta} \log \mathbb{E}_{\hat{p}}[e^{-\eta V}] - \frac{\epsilon}{\eta}$.

## A.2  INVERSE-TEMPERATURE REPARAMETERIZATION AND OPTIMALITY

We directly view $\eta$ as the dual variable. The robust expectation can be expressed as $\inf_{p : D_{\mathrm{KL}}(p \| \hat{p}) \leq \epsilon} \mathbb{E}_p[V] = \sup_{\eta \geq 0} \{ -\frac{1}{\eta} \log \mathbb{E}_{\hat{p}}[e^{-\eta V}] - \frac{\epsilon}{\eta} \}$. The corresponding optimizer is $p_\eta(x) = \frac{\hat{p}(x) e^{-\eta V(x)}}{\mathbb{E}_{\hat{p}}[e^{-\eta V}]}$. Differentiating gives $g'(\eta) = \frac{1}{\eta^2} (\epsilon - D_{\mathrm{KL}}(p_\eta \| \hat{p}))$, so the optimum $\eta^\star$ satisfies $D_{\mathrm{KL}}(p_{\eta^\star} \| \hat{p}) = \epsilon$. Thus $\inf_{p : D_{\mathrm{KL}}(p \| \hat{p}) \leq \epsilon} \mathbb{E}_p[V] = \sup_{\eta \geq 0} \{ -\frac{1}{\eta} \log \mathbb{E}_{\hat{p}}[e^{-\eta V}] - \frac{\epsilon}{\eta} \}$, which provides the derivation of 3.

Instead of enforcing the hard KL constraint, one may consider the soft penalty variant $\inf_p \{ \mathbb{E}_p[V] + \frac{1}{\eta} D_{\mathrm{KL}}(p \| \hat{p}) \}$. The corresponding Lagrangian is $\mathcal{L}(p, \mu) = \sum_x p(x) V(x) + \frac{1}{\eta} \sum_x p(x) (\log p(x) - \log \hat{p}(x)) + \mu (\sum_x p(x) - 1)$ where $\mu$ enforces normalization. Stationarity with respect to $p(x)$ yields $\frac{\partial \mathcal{L}}{\partial p(x)} = V(x) + \frac{1}{\eta} (1 + \log p(x) - \log \hat{p}(x)) + \mu = 0$, so that $p(x) = c \, \hat{p}(x) e^{-\eta V(x)}$. Normalization gives $c^{-1} = Z_\eta = \sum_x \hat{p}(x) e^{-\eta V(x)} = \mathbb{E}_{\hat{p}}[e^{-\eta V}]$ and therefore $p_\eta(x) = \frac{\hat{p}(x) e^{-\eta V(x)}}{Z_\eta}$. Plugging this optimizer into the objective, note that $D_{\mathrm{KL}}(p_\eta \| \hat{p}) = \sum_x p_\eta(x) \log \frac{p_\eta(x)}{\hat{p}(x)} = -\eta \mathbb{E}_{p_\eta}[V] - \log Z_\eta$. Hence $\mathbb{E}_{p_\eta}[V] + \frac{1}{\eta} D_{\mathrm{KL}}(p_\eta \| \hat{p}) = \mathbb{E}_{p_\eta}[V] + \frac{1}{\eta} (-\eta \mathbb{E}_{p_\eta}[V] - \log Z_\eta) = -\frac{1}{\eta} \log Z_\eta$. Since $Z_\eta = \mathbb{E}_{\hat{p}}[e^{-\eta V}]$, we conclude $\inf_p \{ \mathbb{E}_p[V] + \frac{1}{\eta} D_{\mathrm{KL}}(p \| \hat{p}) \} = -\frac{1}{\eta} \log \mathbb{E}_{\hat{p}}[e^{-\eta V}]$.

Thus the soft-penalty form yields the classical entropic risk (risk-sensitive) evaluation with $\eta$ acting as inverse risk-aversion. This variant avoids the explicit $-\epsilon/\eta$ offset of the hard-constrained formulation while preserving robustness via regularization.

## A.3  SAMPLE-BASED APPROXIMATION

In practice, $\hat{p}(\cdot \mid s, a)$ is sampled: $s'_1, \ldots, s'_m \sim \hat{p}$. Sample-based approximations are $\widehat{Z}_\eta = \frac{1}{m} \sum_{i=1}^m e^{-\eta V(s'_i)}$, $\hat{q}_i(\eta) = \frac{e^{-\eta V(s'_i)}}{\sum_j e^{-\eta V(s'_j)}}$, and $\widehat{D}_{\mathrm{KL}}(\eta) = \sum_i \hat{q}_i(\eta) \log(m \hat{q}_i(\eta))$. The approximate dual is $\hat{v}_{\mathrm{rob}}(\eta) = -\frac{1}{\eta} \log \widehat{Z}_\eta - \frac{\epsilon}{\eta}$, and $\eta^\star$ is found by root-finding so that $\widehat{D}_{\mathrm{KL}}(\eta^\star) \approx \epsilon$. Monotonicity holds since $\frac{d}{d\eta} D_{\mathrm{KL}}(p_\eta \| \hat{p}) = \eta \mathrm{Var}_{p_\eta}(V) \geq 0$, enabling bisection or Newton updates. Gradients for critic training follow from $\frac{\partial}{\partial V(s'_i)} \left( -\frac{1}{\eta} \log \widehat{Z}_\eta \right) = \hat{q}_i(\eta)$, so the TD target is $y_t = r_t + \gamma \hat{v}_{\mathrm{rob}}(\eta^\star)$ with backpropagation weight $\gamma \hat{q}_i(\eta^\star)$. By Jensen's inequality, $\mathbb{E}[\log \widehat{Z}_\eta] \leq \log Z_\eta$, so $-\frac{1}{\eta} \log \widehat{Z}_\eta$ is upward biased. At $\eta^\star$, the primal value is $\widehat{\mathbb{E}}_{p_{\eta^\star}}[V] = \sum_i \hat{q}_i(\eta^\star) V(s'_i) = -\frac{\partial}{\partial \eta} \log \widehat{Z}_\eta \big|_{\eta = \eta^\star}$, and ideally coincides with the dual value when $\widehat{D}_{\mathrm{KL}}(\eta^\star) = \epsilon$. In practice small duality gaps may occur, so monitoring $g = \mathrm{dual} - \mathrm{primal}$ is useful.

## A.4  BIAS CORRECTION AND ROOT-FINDING

Since $-\frac{1}{\eta} \log \widehat{Z}_\eta$ is biased upward, concentration yields if $|\widehat{Z}_\eta - Z_\eta| \leq \epsilon Z_\eta$ w.p. $1 - \xi$ then $-\frac{1}{\eta} \log \widehat{Z}_\eta \geq -\frac{1}{\eta} \log Z_\eta - \frac{1}{\eta} \log(1 + \epsilon)$. A conservative estimator is $-\frac{1}{\eta} \log \widehat{Z}_\eta - c_m/\eta$ with

$c_m \simeq \sqrt{\frac{\log(1/\xi)}{m}}$ or bootstrap. Variance may be reduced by larger $m$ or ensembles of dynamics models, with looser $\epsilon$ in high-variance regions. For root-finding, bisection brackets $(0^+, \eta_U)$ by monotonicity; Newton updates use $\eta \leftarrow \eta - \frac{\widehat{D}_{\mathrm{KL}}(\eta) - \epsilon}{\eta \widehat{\mathrm{Var}}_{p_\eta}(V)}$, with projection $\eta > 0$ and damping when variance is small.

### A.5 Entropic Risk Functional, Contraction, and Tractability

The functional $\Psi_\eta(V) = -\frac{1}{\eta} \log \mathbb{E}_{\hat{p}}[e^{-\eta V}]$ enjoys translation invariance $\Psi_\eta(V + c) = \Psi_\eta(V) + c$ and Lipschitz continuity $|\Psi_\eta(V) - \Psi_\eta(W)| \leq \|V - W\|_\infty$, hence the robust Bellman operator $(\mathcal{T}_{\mathrm{KL}} V)(s) = \max_a \{r(s,a) + \gamma \sup_{\eta \geq 0}(\Psi_\eta(V \mid s, a) - \epsilon/\eta)\}$ is a $\gamma$-contraction in $\|\cdot\|_\infty$. Variance control follows from a cumulant expansion: let $\mu = \mathbb{E}_{\hat{p}}[V]$, $U = V - \mu$, and $K_U(t) = \log \mathbb{E}_{\hat{p}}[e^{tU}] = \sum_{n \geq 1} \kappa_n t^n / n!$ be the cumulant generating function of $U$ with $\kappa_1 = 0$, $\kappa_2 = \mathrm{Var}_{\hat{p}}(V)$, etc.; then $\log \mathbb{E}_{\hat{p}}[e^{-\eta V}] = -\eta \mu + K_U(-\eta) = -\eta \mu + \sum_{n \geq 2} \kappa_n \frac{(-\eta)^n}{n!}$, so $\Psi_\eta(V) = \mu - \sum_{n \geq 2} \kappa_n \frac{(-1)^n}{n!} \eta^{n-1} = \mu - \frac{\eta}{2} \mathrm{Var}_{\hat{p}}(V) + \frac{\eta^2}{6} \kappa_3 - \frac{\eta^3}{24} \kappa_4 + O(\eta^4)$, which makes explicit that the first correction to the mean is $-\frac{\eta}{2} \mathrm{Var}_{\hat{p}}(V)$ and higher-order terms depend on higher cumulants, i.e., $\eta$ directly tunes sensitivity to dispersion and tails. In practice, with samples $s'_1, \ldots, s'_m \sim \hat{p}$, one uses $\widehat{\Psi}_{\eta,m}(V) = -\frac{1}{\eta} \log(\frac{1}{m} \sum_{i=1}^m e^{-\eta V(s'_i)})$, which admits concentration bounds of order $\mathcal{O}(e^{2\eta M}/(\eta \sqrt{m}))$ for $V \in [-M, M]$. These properties explain why KL divergence is well-suited in robust Bellman operators: it induces exponential tilting with closed-form optimizers, preserves contraction, and remains tractable while offering explicit variance control via the $\eta$–cumulant expansion.

## B  Boltzmann Reweighting under KL Constraints

We derive the Boltzmann (Gibbs) reweighting that biases an environment ensemble toward adverse models while staying close to a prior. The exponential form follows from the conjugacy of the log-partition function and relative entropy. We give the KL-constrained primal, its dual, the exponential-family solution, and key properties (existence, uniqueness, monotonicity, limits, and the continuous case).

### B.1 Intuitive Motivation: Importance Sampling and Exponential Tilting

We seek a mixture $w$ absolutely continuous w.r.t. a prior $\rho$ to emphasize high vulnerability scores $h_k$ while remaining close to $\rho$. Writing $w(k) = \rho(k) L(k)$ with $L \geq 0$ and $\mathbb{E}_\rho[L] = 1$, an information budget via KL restricts feasible likelihood ratios. Convex duality then yields the optimal $L_\beta(k) \propto e^{-\beta h_k}$ and hence $w_\beta(k) = \frac{\rho(k) e^{-\beta h_k}}{\sum_j \rho(j) e^{-\beta h_j}}$, i.e., exponential tilting (Esscher transform). This guarantees positivity/normalization, aligns with cumulant-generating functions, and is the unique optimizer compatible with a KL constraint, providing a canonical "soft-min" toward adverse models.

### B.2 Setup, Primal, and Dual

Let $\Omega = \{\omega_1, \ldots, \omega_K\}$ index models with prior $\rho \in \Delta_{[K]}$ and scores $h_k \in \mathbb{R}$. We choose $w \in \Delta_{[K]}$ to emphasize adverse models under a KL budget $\kappa \geq 0$:

$$\min_{w \in \Delta_{[K]}} \sum_{k=1}^K w(k) h_k \quad \text{s.t.} \quad D_{\mathrm{KL}}(w \| \rho) \leq \kappa. \tag{6}$$

This is convex and satisfies Slater, so strong duality holds. With multipliers $\lambda \geq 0$ (KL) and $\mu \in \mathbb{R}$ (normalization), the Lagrangian is $\mathcal{L}(w, \lambda, \mu) = \sum_k w(k) h_k + \lambda \left( \sum_k w(k) \log \frac{w(k)}{\rho(k)} - \kappa \right) + \mu \left( \sum_k w(k) - 1 \right)$. Stationarity gives $h_k + \lambda(1 + \log \frac{w(k)}{\rho(k)}) + \mu = 0$, hence $w(k) \propto \rho(k) e^{-h_k/\lambda}$ and after normalization

$$w_\beta^\star(k) = \frac{\rho(k) e^{-\beta h_k}}{\sum_j \rho(j) e^{-\beta h_j}}, \qquad \beta = \tfrac{1}{\lambda} \geq 0. \tag{7}$$

By complementary slackness, the optimal $\beta^\star$ satisfies

$$D_{\mathrm{KL}}(w_{\beta^\star}^\star \| \rho) = \kappa. \tag{8}$$

Minimizing over $w$ yields the dual function $g(\lambda) = -\lambda \log \sum_k \rho(k) e^{-h_k/\lambda} - \lambda\kappa$, equivalently

$$\max_{\beta \geq 0} \left\{ -\frac{1}{\beta} \log \sum_{k=1}^K \rho(k) e^{-\beta h_k} - \frac{\kappa}{\beta} \right\}, \tag{9}$$

whose maximizer $\beta^\star$ together with 7 solves 6. The term $-(1/\beta) \log \sum_k \rho(k) e^{-\beta h_k}$ is the negative free energy (Gibbs principle).

### B.3 CONVEX-ANALYTIC VIEW (DONSKER–VARADHAN)

For any $f = (f_k)$, $\log \sum_k \rho(k) e^{f_k} = \sup_{w \in \Delta_{[K]}} \{ \sum_k w(k) f_k - D_{\mathrm{KL}}(w \| \rho) \}$. Setting $f_k = -\beta h_k$ gives $-\frac{1}{\beta} \log \sum_k \rho(k) e^{-\beta h_k} = \inf_{w \in \Delta_{[K]}} \{ \sum_k w(k) h_k + \frac{1}{\beta} D_{\mathrm{KL}}(w \| \rho) \}$. Introducing the budget $\kappa$ via a Lagrange multiplier recovers 9 and 7. Exponential tilting is thus the optimizer of an entropy-regularized worst-case objective.

### B.4 PROPERTIES, LIMITS, AND CONTINUOUS CASE

*Existence/uniqueness:* strong duality holds by Slater. When $h_k$ are not all equal and $\kappa > 0$, 8 has a unique solution $\beta^\star$, hence a unique $w_{\beta^\star}^\star$. *Monotonicity:* with $Z(\beta) = \sum_k \rho(k) e^{-\beta h_k}$ and $w_\beta$ as in 7, $\frac{d}{d\beta} D_{\mathrm{KL}}(w_\beta \| \rho) = \beta \operatorname{Var}_{w_\beta}(h) \geq 0$, with strict inequality unless $h$ is $\rho$-a.s. constant. Continuity and strict increase guarantee a unique $\beta^\star$ matching $\kappa$. *Temperature limits:* as $\beta \to 0$, $w_\beta \to \rho$. As $\beta \to \infty$, $w_\beta$ concentrates on $\arg\min_k h_k$, turning the soft-min into a hard minimum. *Continuous ensembles:* on $(\Omega, \mathcal{F})$ with prior $\rho$ and measurable $h : \Omega \to \mathbb{R}$, $\frac{dw_\beta}{d\rho}(\omega) = \frac{e^{-\beta h(\omega)}}{\int_\Omega e^{-\beta h(\tilde{\omega})} \rho(d\tilde{\omega})}$, with $D_{\mathrm{KL}}(w_{\beta^\star} \| \rho) = \kappa$, obtained by replacing sums with integrals.

Starting from 6, Lagrangian duality or the Gibbs variational principle yields the exponential-family optimizer 7, with inverse temperature $\beta$ trading off energy (vulnerability) and entropy (deviation from prior) and chosen by 8. This is the Boltzmann reweighting used to bias sampling toward adverse regimes in a principled, tunable way.

### B.5 WHY KL CONSTRAINT IN BOLTZMANN REWEIGHTING

For an ensemble $\Omega = \{\omega_1, \ldots, \omega_K\}$ with prior $\rho \in \Delta_{[K]}$ and vulnerability scores $h_k \in \mathbb{R}$, the KL-constrained reweighting problem

$$\min_{w \in \Delta_{[K]}} \sum_{k=1}^K w(k) h_k \quad \text{s.t.} \quad D_{\mathrm{KL}}(w \| \rho) \leq \kappa \tag{10}$$

admits the Boltzmann solution $w_\beta(k) = \frac{\rho(k) e^{-\beta h_k}}{\sum_j \rho(j) e^{-\beta h_j}}$ with $D_{\mathrm{KL}}(w_\beta \| \rho) = \kappa$, which arises from strong duality and exponential tilting. This closed-form structure is unique to KL divergence: other $f$-divergences yield polynomial or truncated forms such as $w(k) \propto \rho(k)(1 - \beta h_k)_+$ for $\chi^2$, which lack smoothness and global stability. The mapping $\beta \mapsto D_{\mathrm{KL}}(w_\beta \| \rho)$ is continuous, strictly increasing, and unbounded unless all $h_k$ coincide, guaranteeing that for each finite $\kappa > 0$ there exists a unique $\beta^\star$ satisfying the constraint. Thus $w_\beta$ interpolates smoothly between $w = \rho$ at $\beta = 0$ (no reweighting) and the hard-min distribution as $\beta \to \infty$. Finally, the budget $\kappa$ has a direct information-theoretic meaning: $D_{\mathrm{KL}}(w \| \rho) \leq \kappa$ bounds the expected log-likelihood ratio $\mathbb{E}_w[\log(w/\rho)]$, so $\kappa$ quantifies the exact information distance from the prior and explicitly limits how much prior mass can be shifted toward adverse models.

## C ON THE USE OF ADVNET FOR $\eta$ APPROXIMATION

In Sec. 4, we introduced AdvNet as a practical mechanism to approximate the dual variable $\eta^\star$ that enforces the KL budget in RAPO. Here we provide further theoretical background and justification: we first explain why the hard minimum is infeasible, then review the exact dual optimization

and the interpretation of $\eta$ as a KL–sharpness trade-off, and finally analyze AdvNet's amortized approximation, including its stability and error propagation properties.

## C.1   WHY NOT SIMPLY TAKE THE MINIMUM?

A natural question is whether one can avoid the dual altogether and simply take the hard minimum over samples: $\min_i V(s'_i)$ with $s'_i \sim \hat{p}(\cdot|s,a)$. This corresponds to choosing the degenerate distribution $q = \delta_{s'_{\min}}$. However, $D_{\mathrm{KL}}(\delta_{s'_{\min}} \| \hat{p}) = \log \frac{1}{\hat{p}(s'_{\min})}$, which equals $\log m$ if $\hat{p}$ is approximated by a uniform empirical distribution over $m$ samples. Unless $\epsilon \geq \log m$, the hard minimum violates the KL constraint and is therefore infeasible. In continuous spaces it becomes even worse: concentrating on a single point where $\hat{p}(s') = 0$ yields infinite KL.

Thus, the true robust solution under a finite KL budget is the *soft minimum* distribution $q_\eta$ defined by exponential tilting. The concentration of $q_\eta$ is controlled by $\eta$, which is uniquely chosen so that the KL budget is exactly satisfied.

## C.2   DUAL OPTIMIZATION VERSUS ADVNET APPROXIMATION

The dual variable $\eta^\star$ acts as the inverse temperature that enforces the KL constraint in the tilted mixture. In principle, one can solve a one-dimensional convex optimization for each $(s,a)$: $\eta^\star(s,a) = \arg\max_{\eta \geq 0}\{-\frac{1}{\eta}\log \mathbb{E}_{\hat{p}(\cdot|s,a)}[e^{-\eta V(s')}] - \frac{\epsilon}{\eta}\}$, and then compute the reweighted distribution $q_{\eta^\star}(s') = \frac{\hat{p}(s'|s,a)e^{-\eta^\star V(s')}}{\mathbb{E}_{\hat{p}}[e^{-\eta^\star V(s')}]}$ with $D_{\mathrm{KL}}(q_{\eta^\star}\|\hat{p}) = \epsilon$. This dual root-finding is exact but requires iterative optimization for every state-action query.

AdvNet is introduced as an *amortized approximation*: rather than solving for $\eta^\star$ repeatedly, we train a neural network $\hat{\eta} = f_\phi(s,a)$ to approximate the mapping $(s,a) \mapsto \eta^\star$. Training is supervised against either the dual solution $\eta^\star$ or the induced robust surrogate loss. Once trained, AdvNet provides a fast forward-pass approximation for arbitrary $(s,a)$.

## C.3   INTERPRETATION OF $\eta$ AND THE KL–SHARPNESS TRADE-OFF

The tilting distribution $q_\eta(s') = \frac{\hat{p}(s'|s,a)e^{-\eta V(s')}}{\mathbb{E}_{\hat{p}}[e^{-\eta V(s')}]}$ places more mass on adverse states as $\eta$ increases. Consequently, $D_{\mathrm{KL}}(q_\eta\|\hat{p})$ is strictly increasing in $\eta$. At $\eta = 0$, $q_\eta = \hat{p}$ and KL$= 0$. As $\eta \to \infty$, $q_\eta$ collapses to the worst-case delta distribution and KL diverges. Hence $\eta^\star$ is the sharpness level that exactly matches the KL budget.

Large $\eta$ is not always better: while it yields stronger concentration on the worst case, it also violates the KL constraint and produces unstable gradients. The robust expectation balances both effects through the dual objective $\sup_{\eta \geq 0}\{-\frac{1}{\eta}\log \mathbb{E}_{\hat{p}}[e^{-\eta V}] - \frac{\epsilon}{\eta}\}$.

## C.4   WHY THE DUAL TEMPERATURE $\eta$ IS STATE–ACTION DEPENDENT

We show that the optimal dual temperature arises *separately at each state–action pair* from the hard KL-constrained inner problem, and under mild regularity it is a well-defined (locally smooth) function of $(s,a)$.

**Setup: one-step robust evaluation at $(s,a)$.**   Fix a policy $\pi$ and value function $V$. With $(s,a)$-rectangular KL balls $\mathcal{P}_\epsilon(s,a) = \{p(\cdot \mid s,a) \in \Delta_\mathcal{S} : D_{\mathrm{KL}}(p\|\hat{p}(\cdot \mid s,a)) \leq \epsilon\}$, the robust Bellman operator (Prelim., 2) requires, *for each $(s,a)$*, the one-step inner minimum $v_{\mathrm{rob}}(s,a) = \inf_{p \in \mathcal{P}_\epsilon(s,a)} \mathbb{E}_{s'\sim p(\cdot|s,a)}V(s')$. By convex duality for the hard KL constraint, this is equivalently $v_{\mathrm{rob}}(s,a) = \sup_{\eta \geq 0}\{-\frac{1}{\eta}\log \mathbb{E}_{\hat{p}(\cdot|s,a)}[e^{-\eta V(s')}] - \frac{\epsilon}{\eta}\} = \sup_{\eta \geq 0} \mathcal{L}(s,a;\eta)$. Thus the dual variable $\eta$ is the Lagrange multiplier of a *local* constraint at $(s,a)$.

**First-order condition and tightness of the KL budget.**   Write $Z_{s,a}(\eta) = \mathbb{E}_{\hat{p}(\cdot|s,a)}[e^{-\eta V(s')}]$ and $q_{s,a,\eta}(s') = \frac{e^{-\eta V(s')}}{Z_{s,a}(\eta)}\hat{p}(s' \mid s,a)$. Then $D_{\mathrm{KL}}(q_{s,a,\eta} \| \hat{p}(\cdot \mid s,a)) = -\eta\,\mathbb{E}_{q_{s,a,\eta}}[V] - \log Z_{s,a}(\eta)$.

A standard calculation yields $\frac{\partial \mathcal{L}}{\partial \eta}(s, a; \eta) = \frac{\epsilon - D_{\mathrm{KL}}(q_{s,a,\eta} \| \hat{p}(\cdot | s,a))}{\eta^2}$. Hence any interior maximizer $\eta^\star(s, a)$ satisfies the tightness condition $D_{\mathrm{KL}}(q_{s,a,\eta^\star(s,a)} \| \hat{p}(\cdot \mid s, a)) = \epsilon$.

**Monotonicity, unimodality, and uniqueness.** For $\eta > 0$, $\frac{d}{d\eta} D_{\mathrm{KL}}(q_{s,a,\eta} \| \hat{p}(\cdot \mid s, a)) = \eta \operatorname{Var}_{q_{s,a,\eta}}[V(s')] \geq 0$. Unless $V$ is $q_{s,a,\eta}$-a.s. constant, the derivative is positive, so $\eta \mapsto D_{\mathrm{KL}}(q_{s,a,\eta} \| \hat{p})$ is strictly increasing. Combining with the gradient expression, $\eta \mapsto \mathcal{L}(s, a; \eta)$ is unimodal with a *unique* maximizer where the tightness condition holds. Therefore, for each $(s, a)$, the optimal dual temperature $\eta^\star(s, a) = \arg\max_{\eta \geq 0} \mathcal{L}(s, a; \eta)$ is uniquely defined (up to boundary cases). This shows $\eta^\star$ is *inherently $(s, a)$-dependent*: the data of the local problem—$\hat{p}(\cdot \mid s, a)$ and the induced distribution of $V(s')$—change with $(s, a)$, and so does the unique maximizer.

**Regularity: when does $\eta^\star(s, a)$ behave like a function?** Define the zero set $\Phi(s, a, \eta) = D_{\mathrm{KL}}(q_{s,a,\eta} \| \hat{p}(\cdot \mid s, a)) - \epsilon$. Under standard continuity assumptions in $(s, a)$ for $\hat{p}(\cdot \mid s, a)$ and $V$, $\Phi$ is continuous and $\mathcal{C}^1$ in $\eta$. Moreover, $\frac{\partial \Phi}{\partial \eta}(s, a, \eta) = \eta \operatorname{Var}_{q_{s,a,\eta}}[V] > 0$ whenever $\eta > 0$ and the variance is nonzero. Hence, by the Implicit Function Theorem, any interior solution $\eta^\star(s, a)$ of $\Phi = 0$ defines a *locally $\mathcal{C}^1$* function of $(s, a)$. Furthermore, since $\mathcal{L}(s, a; \eta)$ is continuous in $(s, a, \eta)$ and strictly concave in $\eta$, Berge's Maximum Theorem implies that the argmax correspondence is single-valued and continuous in $(s, a)$. In sum, under mild non-degeneracy (nonzero variance) and continuity, we may treat $\eta^\star$ as a well-defined, locally smooth *function* $(s, a) \mapsto \eta^\star(s, a)$.

**Boundary and degenerate cases.** If $\operatorname{Var}_{q_{s,a,\eta}}[V] = 0$ on an interval, or if $\epsilon$ is so small/large that the solution lies at the boundary, then $\eta^\star(s, a)$ may occur at $\eta = 0$ or at a large cutoff. In practice we project $\eta$ to $[\eta_{\min}, \eta_{\max}]$ and (if needed) use one-dimensional bisection enforcing the tightness condition.

**Implication for RAPO (AdvNet).** Because the inner problem is solved *for each* $(s, a)$, the robust Bellman operator requires the *local* temperature $\eta^\star(s, a)$. RAPO's AdvNet amortizes this implicit solution by learning a predictor $f_\psi(s, a) \approx \eta^\star(s, a)$, which is not a heuristic convenience but a direct realization of the dual structure: it supplies the state–action dependent temperature that makes the inner KL constraint tight and yields the correct soft-min target $-\frac{1}{\eta} \log \mathbb{E}_{\hat{p}}[e^{-\eta V}]$ at each $(s, a)$.

## C.5 Amortized Approximation via AdvNet: Motivation, Stability, and Error Analysis

The term "amortized" follows variational inference: instead of optimizing a local variable $\eta^\star$ for each instance, we train a global function $f_\psi(s, a)$ that amortizes the inference cost. The mapping $(s, a) \mapsto \eta$ may seem unintuitive, but is learnable in practice because different regions of the state-action space consistently correlate with different adverse dynamics.

*Example.* Consider a pendulum with uncertain mass. For the same $(s, a)$, a lighter pendulum responds faster, while a heavier pendulum is sluggish. If the policy applies large torques, heavy mass instances are more failure-prone. AdvNet can learn this regularity, outputting larger $\eta$ for such $(s, a)$ pairs to bias reweighting toward heavier models.

To summarize, the dual variable $\eta^\star$ can in principle be computed exactly by solving the dual optimization, but this is computationally expensive; AdvNet provides an amortized approximation that replaces repeated per-sample root-finding with a single learned forward pass. Since hard minima correspond to degenerate distributions and are infeasible under finite KL budgets, the dual soft-minimum defined by $\eta^\star$ is the correct solution. The parameter $\eta$ mediates the trade-off between worst-case sharpness and divergence from the nominal prior, and AdvNet offers an efficient practical mechanism to approximate this trade-off.

We now formalize the role of the state–action dependent dual variable. For a fixed $(s, a)$ with nominal kernel $\hat{p}(\cdot \mid s, a)$ and a bounded $V : \mathcal{S} \to \mathbb{R}$, the dual representation of the one-step robust evaluation under a hard KL constraint is $\phi(\eta; V) = -\frac{1}{\eta} \log \mathbb{E}_{\hat{p}}[e^{-\eta V}] - \frac{\epsilon}{\eta}$, $\eta \geq 0$, and the maximizer $\eta^\star(s, a) \in \arg\max_{\eta \geq 0} \phi(\eta; V \mid s, a)$ exists uniquely. By the KKT conditions, $D_{\mathrm{KL}}(q_{\eta^\star} \| \hat{p}) = \epsilon$ and $q_\eta(\mathrm{d}s') \propto e^{-\eta V(s')} \hat{p}(\mathrm{d}s')$. Since $V(\cdot \mid s, a)$ varies with $(s, a)$, the optimizer $\eta^\star$ is inherently state–action dependent.

**Contraction is preserved irrespective of AdvNet approximation.** Fix any measurable mapping $\hat{\eta} : \mathcal{S} \times \mathcal{A} \to \mathbb{R}_{\geq 0}$ and define $(\widetilde{\mathcal{T}}V)(s) = \max_a \{r(s,a) + \gamma(\Psi_{\hat{\eta}(s,a)}(V \mid s,a) - \epsilon/\hat{\eta}(s,a))\}$ with $\Psi_\eta(V) = -\frac{1}{\eta}\log\mathbb{E}_{\hat{p}}[e^{-\eta V}]$. For all $\eta > 0$, $\Psi_\eta$ is 1-Lipschitz in $\|\cdot\|_\infty$, i.e., $|\Psi_\eta(V) - \Psi_\eta(W)| \leq \|V - W\|_\infty$. Thus, for arbitrary $\hat{\eta}(\cdot)$, $\|\widetilde{\mathcal{T}}V - \widetilde{\mathcal{T}}W\|_\infty \leq \gamma\|V - W\|_\infty$, so $\widetilde{\mathcal{T}}$ remains a $\gamma$-contraction. Hence, even if AdvNet does not converge exactly to $\eta^\star$, the operator still converges to a unique fixed point without loss of contraction.

**Strong concavity of the dual objective and quadratic error effects.** The derivative of $\phi$ is $\phi'(\eta) = \frac{1}{\eta^2}(\epsilon - D_{\mathrm{KL}}(q_\eta\|\hat{p}))$, and $\phi''(\eta) = -\frac{2}{\eta^3}(\epsilon - D_{\mathrm{KL}}(q_\eta\|\hat{p})) - \frac{1}{\eta}\mathrm{Var}_{q_\eta}(V) \leq 0$. Thus $\phi$ is strictly concave, ensuring a unique optimum $\eta^\star$. By Taylor expansion, $\phi(\eta^\star) - \phi(\hat{\eta}) = -\frac{1}{2}\phi''(\xi)(\hat{\eta} - \eta^\star)^2 \leq \frac{1}{2}\frac{\mathrm{Var}_{q_\xi}(V)}{\xi}(\hat{\eta} - \eta^\star)^2$. If $V \in [m, M]$, then $\mathrm{Var}_{q_\xi}(V) \leq (M-m)^2/4$. Restricting $\eta \geq \eta_{\min} > 0$ yields $0 \leq \phi(\eta^\star) - \phi(\hat{\eta}) \leq \frac{(M-m)^2}{8\,\eta_{\min}}(\hat{\eta} - \eta^\star)^2$. Hence the dual value loss grows only quadratically with the approximation error.

**KL residual and local equivalence.** Optimality satisfies $r(\eta) := \epsilon - D_{\mathrm{KL}}(q_\eta\|\hat{p}) = 0$. Since $r'(\eta) = -\eta\mathrm{Var}_{q_\eta}(V) \leq -\eta_{\min}\sigma_{\min}^2$ with variance lower bound $\sigma_{\min}^2 > 0$, the inverse function theorem implies $|\hat{\eta} - \eta^\star| \leq \frac{|\epsilon - D_{\mathrm{KL}}(q_{\hat{\eta}}\|\hat{p})|}{\eta_{\min}\sigma_{\min}^2}$. Substituting into the quadratic bound, $\phi(\eta^\star) - \phi(\hat{\eta}) \leq \frac{(M-m)^2}{8\,\eta_{\min}^3\sigma_{\min}^4}\big(\epsilon - D_{\mathrm{KL}}(q_{\hat{\eta}}\|\hat{p})\big)^2$. Thus penalizing or projecting KL residuals, as in our training objective, directly controls approximation error.

**Operator-level performance gap.** Let $(\mathcal{T}_{\mathrm{KL}}V)(s) = \max_a \sup_{\eta\geq 0}\{r + \gamma(\Psi_\eta(V) - \epsilon/\eta)\}$ denote the exact operator, and $\widetilde{\mathcal{T}}$ the approximate one. For any $V$, $|(\mathcal{T}_{\mathrm{KL}}V)(s) - (\widetilde{\mathcal{T}}V)(s)| \leq \gamma\max_a(\phi(\eta^\star) - \phi(\hat{\eta}(s,a))) \leq \gamma C_\phi\max_a|\hat{\eta}(s,a) - \eta^\star(s,a)|^2$, with $C_\phi = (M-m)^2/(8\,\eta_{\min})$. By contraction perturbation bounds, the fixed points $V^\star$ and $\tilde{V}$ satisfy $\|V^\star - \tilde{V}\|_\infty \leq \frac{\gamma}{1-\gamma}C_\phi\sup_{s,a}|\hat{\eta}(s,a) - \eta^\star(s,a)|^2$. Hence, if AdvNet's output error is uniformly bounded by $\bar{\delta}$, the resulting value function deviation is at most $O(\delta^2)$.

**Summary.** (i) The optimal $\eta^\star$ is inherently state–action dependent, motivating AdvNet as a natural approximation. (ii) Contraction of the robust Bellman operator is preserved for any approximate $\hat{\eta}$. (iii) Approximation error in $\eta$ only induces a *quadratic* gap in the robust value. (iv) KL residuals provide a direct and quantifiable proxy for approximation accuracy. Together, these results rigorously justify the use of AdvNet: even without exact alignment to $\eta^\star$, convergence is preserved and approximation error propagates only as a second-order effect.

# D    EXPERIMENT DETAILS AND ADDITIONAL RESULTS

## D.1    ROBUSTNESS TO DYNAMICS UNCERTAINTY

We complement the `Walker2d` results in the main text with robustness sweeps on `Ant`, `Halfcheetah`, and `Hopper`. Across all environments RAPO consistently attains the best trade-off between ID performance and OOD stability, while maintaining the strict ordering RAPO > PPO+DR > PPO that we already observed in `Walker2d`. The relative gaps between algorithms vary with morphology. For `Ant`, the curves are closer overall, yet RAPO still sustains high returns at OOD scales where baselines collapse. `Halfcheetah` shows sharper degradation of PPO once parameters move outside the training distribution, while PPO+DR delays the collapse but remains vulnerable; RAPO keeps returns high with only mild declines. `Hopper` is the most sensitive to dynamics changes: PPO and RARL both degrade quickly in OOD, EPOpt is conservative but stable, and RAPO preserves performance substantially better than all baselines.

These experiments confirm several observations made on `Walker2d`. First, robustness does not come for free: RAPO evaluates multiple dynamics models per update and applies Boltzmann reweighting, so although rollout sample counts are fixed, per-step compute is higher than PPO. Second, Boltzmann reweighting concentrates learning on difficult parameter regimes, which produces informative non-monotonic effects such as inertia scale 0.5 outperforming 0.6 or torque showing

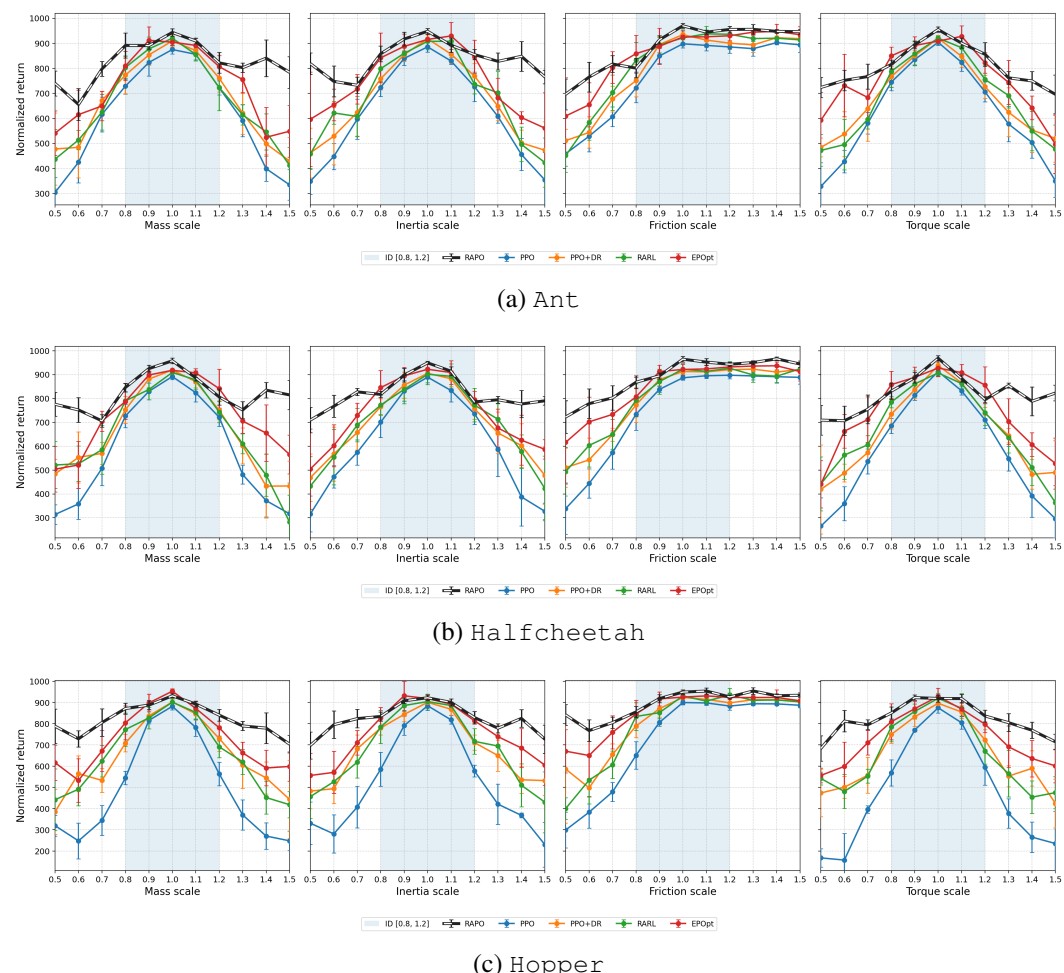

(a) `Ant`

(b) `Halfcheetah`

(c) `Hopper`

Figure 6: Robustness curves across mass, inertia, friction, and torque scaling. RAPO consistently outperforms baselines in OOD regions while retaining strong ID performance.

left-OOD gains when the sampler emphasizes challenging configurations. Third, friction is effectively flat for scales $\geq 1.0$ across all environments. Finally, even under uniform domain randomization, ID-region returns are not uniform: performance decreases toward the boundaries of the prior, underscoring that uniform sampling does not equal uniform task difficulty.

### D.2 ROLE OF ADVNET AND BOLTZMANN REWEIGHTING

We extend the ablation study beyond `Walker2d` by evaluating RAPO and its variants on `Ant`, `Halfcheetah`, and `Hopper`. In all cases, training budgets and evaluation protocols are identical; the only differences are whether the trajectory adversary (AdvNet) and/or Boltzmann reweighting are active. Full RAPO consistently achieves the most stable curves in both ID and OOD regions. Removing one mechanism reduces robustness, and removing both produces the steepest OOD collapse, showing that trajectory stress testing and tilted sampling are complementary.

*RAPO without AdvNet* disables the adversarial head and sets its loss weight to zero, while retaining Boltzmann tilting. For each update the dual temperature $\eta^\star$ is obtained via convex dual root finding so that the tilted weights $w_k = \frac{\rho_k \exp(\eta^\star \Delta_k)}{\sum_j \rho_j \exp(\eta^\star \Delta_j)}$ satisfy the KL constraint, where $\Delta_k$ are Monte Carlo critic shortfalls. This variant captures globally adverse dynamics but misses trajectory-specific fragilities, leading to wider confidence intervals and occasional local dips. *RAPO without Boltzmann reweighting* fixes $w = \rho$, i.e., the sampler matches the prior distribution over dynamics, while AdvNet continues to perturb trajectories. This variant maintains mid-range OOD competitiveness

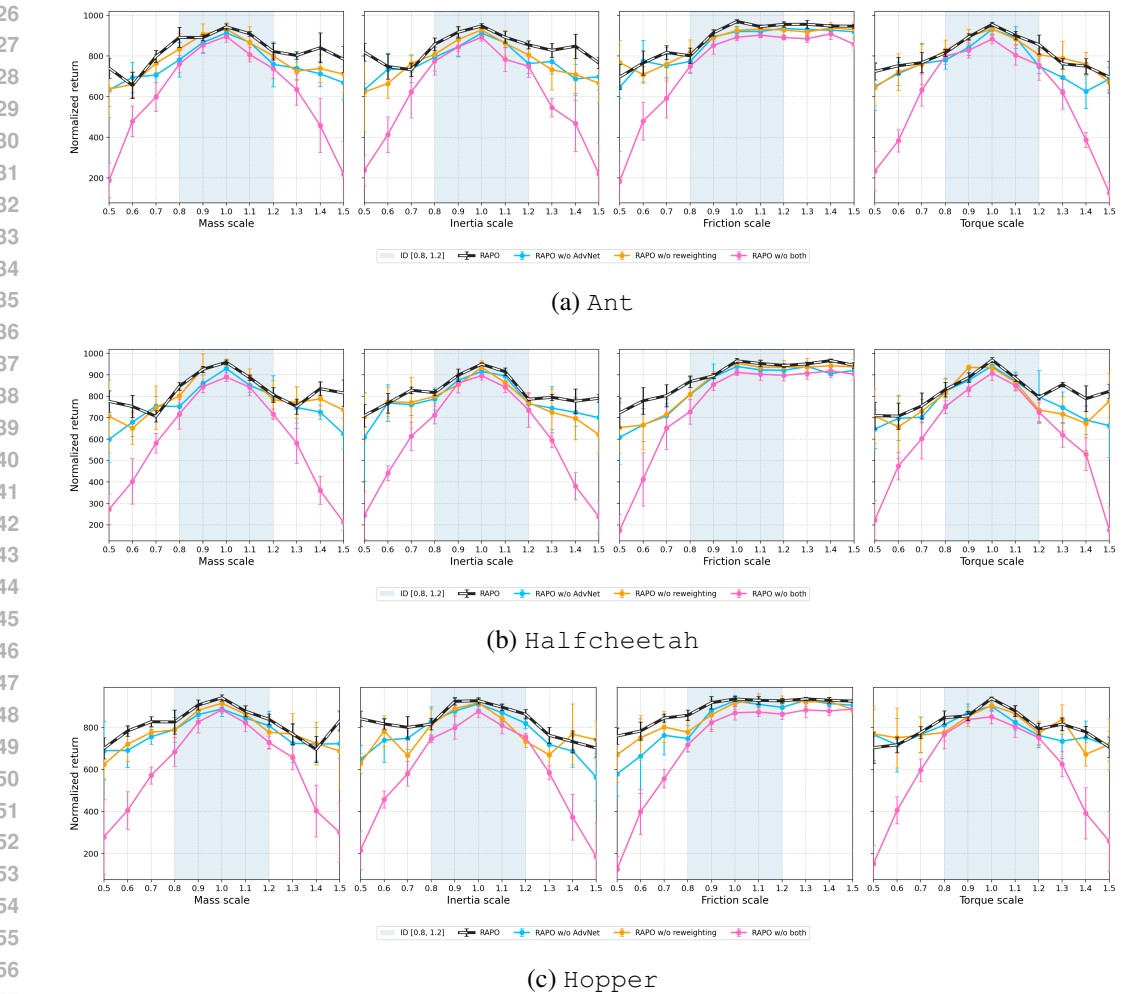

(a) `Ant`

(b) `Halfcheetah`

(c) `Hopper`

Figure 7: Ablation of RAPO components on `Ant`, `Halfcheetah`, and `Hopper`. Full RAPO achieves the best OOD robustness. Removing AdvNet or Boltzmann reweighting weakens robustness, and removing both leads to the sharpest collapse.

but decays symmetrically at extremes because harmful models are undersampled. *RAPO without both* fixes $w = \rho$ and removes AdvNet, effectively reducing training to PPO on the un-tilted mixture, which results in brittle tails centered near the nominal scale.

Across environments the patterns are consistent with `Walker2d`. In `Ant`, RAPO sustains the highest OOD returns, while ablations fall off more sharply. In `Halfcheetah`, PPO-based variants degrade rapidly once parameters move OOD, but RAPO without AdvNet or without reweighting still preserve partial robustness; the combined removal collapses fastest. `Hopper` is especially sensitive to dynamics changes: RAPO provides stable OOD returns, whereas the ablations show steeper declines, particularly when both mechanisms are disabled. These results highlight that AdvNet and Boltzmann reweighting address different aspects of robustness—state–action dependent fragility versus global coverage—and that both are needed for the strongest performance.

To further illustrate robustness, Fig. 8 reports value-function heatmaps for `Ant`, `Halfcheetah`, and `Hopper`. For each update $u$ and parameter type, we sweep a dense grid of scales $\alpha \in [0.5, 1.5]$ while holding the other parameters at 1.0, and compute $V_u(\alpha)$ from one-step critic rollouts. Concretely, given a batch of states $\{s_b\}_{b=1}^{B}$ sampled from the replay buffer, we evaluate actions $a_b \sim \pi_u(\cdot|s_b)$, step the simulator with parameter scale $\alpha$ to obtain $s_b'$, and form $V_u(\alpha) = \frac{1}{B} \sum_{b=1}^{B} \left( r(s_b, a_b; \alpha) + \gamma \hat{V}_{\phi_u}(s_b') \right)$, where $\hat{V}_{\phi_u}$ is the learned critic at update $u$. This

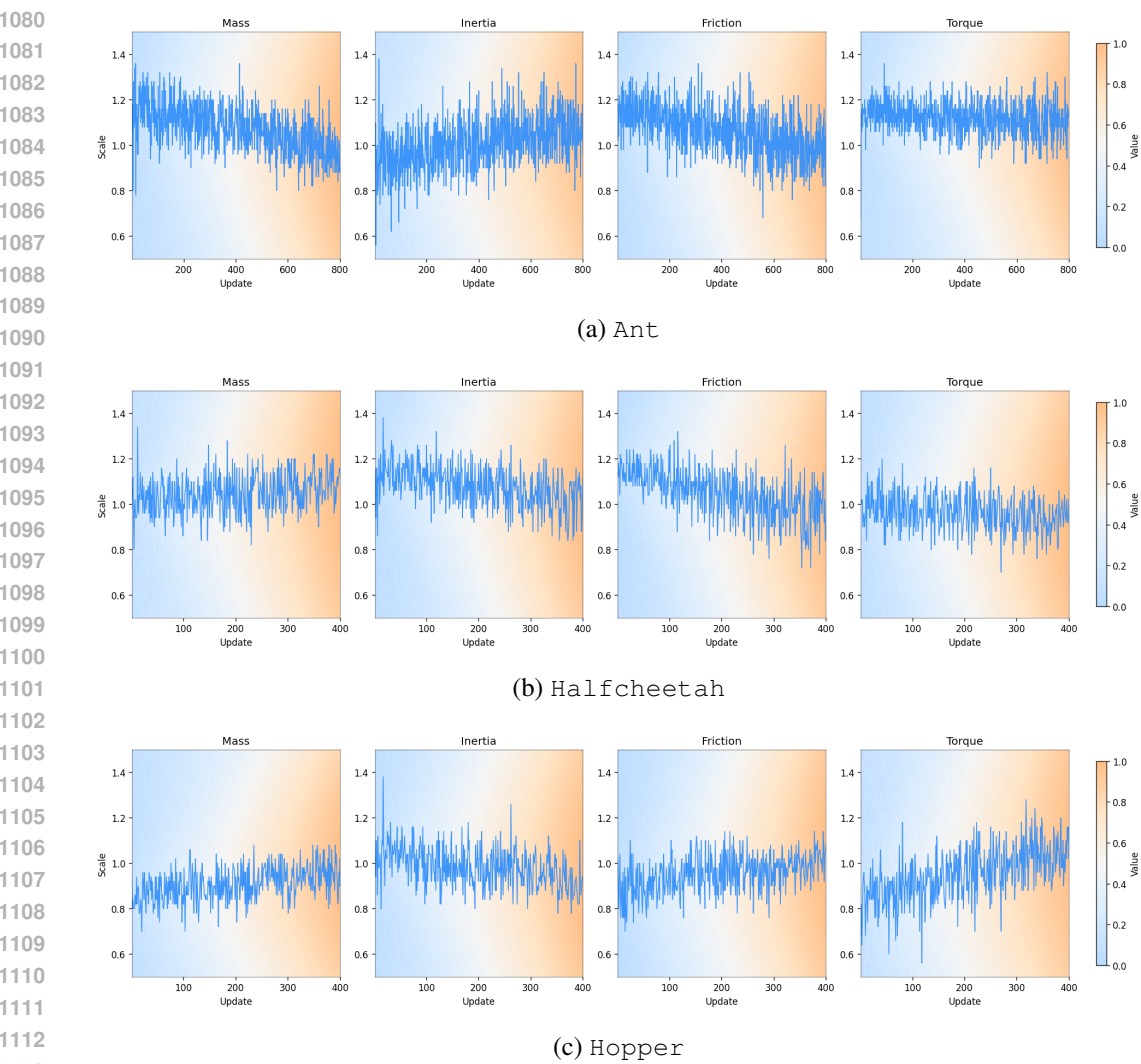

(a) Ant

(b) Halfcheetah

(c) Hopper

Figure 8: Heatmaps of value estimates with and without Boltzmann reweighting. RAPO progressively lifts the value floor across the entire range of scales, while removing reweighting leaves OOD tails dark, indicating poor robustness.

diagnostic is continuous even though RAPO training uses a finite $K$-mixture. Brighter colors correspond to higher values, and the blue trace marks $\arg\max_\alpha V_u(\alpha)$.

Across all three environments, RAPO gradually raises the value floor over the entire scale range. Early updates exhibit dark bands at extreme OOD scales, which flatten and brighten as training progresses. Without Boltzmann reweighting ($w = \rho$), value remains concentrated around ID scales while OOD tails stay dark, reflecting poor coverage of adverse dynamics.

To complement the main-text heatmaps, we provide additional results for the variant without Boltzmann reweighting ($w = \rho$). In this setting, value estimates remain concentrated near ID scales while OOD tails stay persistently dark, confirming that Boltzmann tilting is essential for coverage of adverse dynamics. Unlike full RAPO, which gradually raises the value floor across the full parameter range, the no-reweighting variant leaves hard regimes under-explored and fragile.

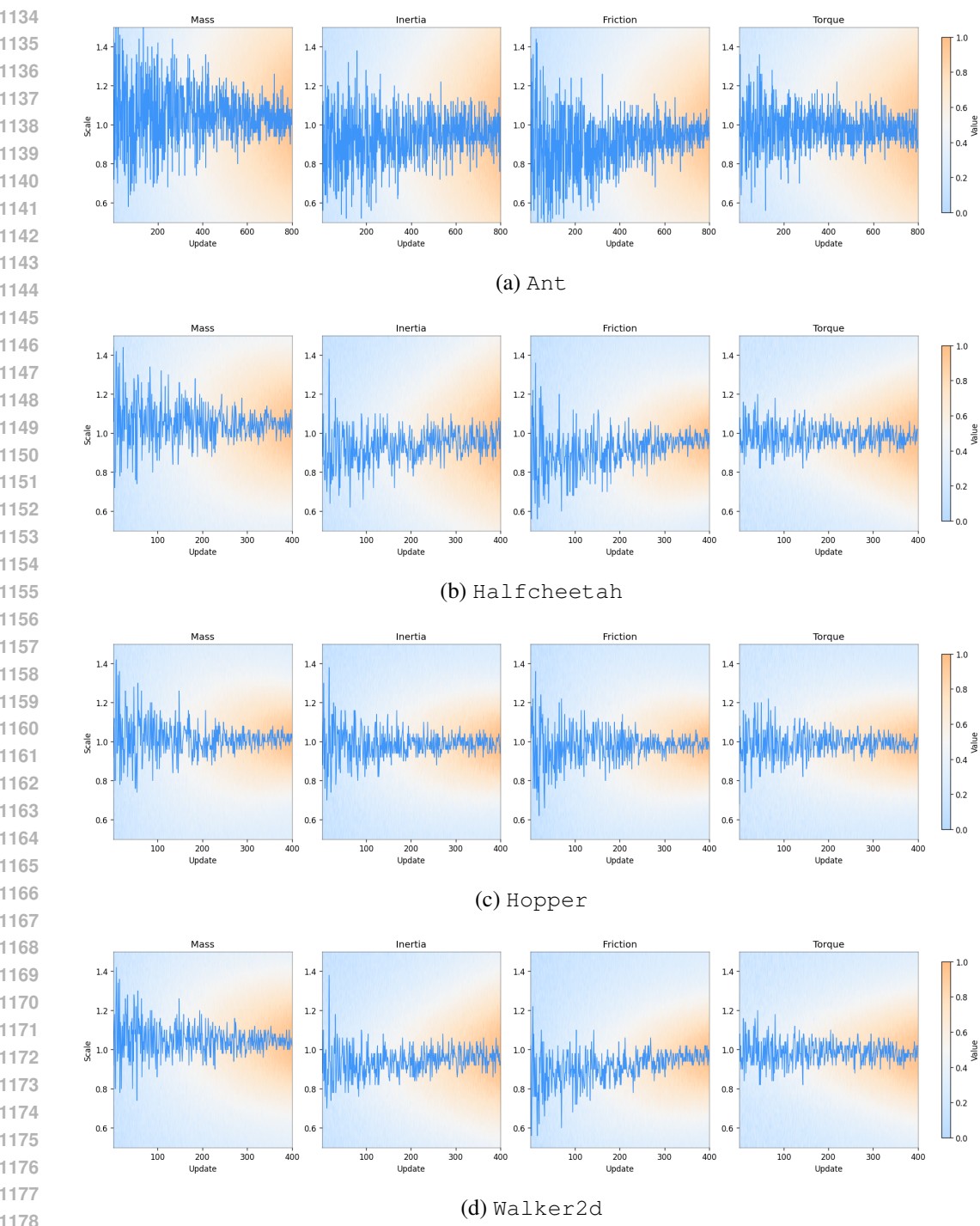

Figure 9: Heatmaps of value estimates without Boltzmann reweighting ($w = \rho$). Compared to the full RAPO results in the main text, bright regions remain confined to ID scales and OOD tails stay dark, reflecting poor coverage of challenging dynamics.

Table 2: Sensitivity of RAPO to ensemble size $K$ (top) and Monte Carlo samples $m$ (bottom). Performance is normalized return (mean $\pm$ std over 5 seeds). All runs finish within $\leq 45$ min.

| Ensemble size $K$ sensitivity ($m = 16$) | | | | |
|---|---|---|---|---|
| Env | $K = 4$ | $K = 8$ | $K = 16$ | $K = 32$ |
| Hopper | $0.72 \pm .03$ | $0.76 \pm .02$ | $0.78 \pm .02$ | $0.77 \pm .03$ |
| Walker2d | $0.65 \pm .06$ | $0.70 \pm .04$ | $0.73 \pm .05$ | $0.75 \pm .05$ |
| HalfCheetah | $0.60 \pm .05$ | $0.65 \pm .04$ | $0.67 \pm .03$ | $0.67 \pm .04$ |
| Ant | $0.50 \pm .06$ | $0.58 \pm .04$ | $0.62 \pm .03$ | $0.61 \pm .03$ |
| Monte Carlo samples $m$ sensitivity ($K = 16$) | | | | |
| Env | $m = 4$ | $m = 8$ | $m = 16$ | $m = 32$ |
| Hopper | $0.74 \pm .05$ | $0.76 \pm .03$ | $0.77 \pm .02$ | $0.77 \pm .02$ |
| Walker2d | $0.68 \pm .06$ | $0.70 \pm .04$ | $0.72 \pm .03$ | $0.72 \pm .03$ |
| HalfCheetah | $0.62 \pm .05$ | $0.65 \pm .04$ | $0.66 \pm .03$ | $0.66 \pm .03$ |
| Ant | $0.55 \pm .07$ | $0.58 \pm .05$ | $0.60 \pm .03$ | $0.60 \pm .03$ |

| Parameters | Values |
|---|---|
| Mass | 0.280 kg |
| Inertia around $x, y$ | $2.36 \times 10^{-4}$ kg $\cdot$ m$^2$ |
| Inertia around $z$ | $3.03 \times 10^{-4}$ kg $\cdot$ m$^2$ |
| Arm length | 0.058 m |
| Propeller thrust factor | $1.145 \times 10^{-7}$ N $\cdot$ s$^2$ |
| Payload mass | 0.025 kg |

Table 3: Nominal quadrotor parameters.

### D.3 SENSITIVITY TO ENSEMBLE SIZE $K$ AND MONTE CARLO SAMPLES $m$

### D.4 QUADROTOR SIMULATION DETAILS

**Simulation Platform.** All evaluations are conducted in high-fidelity simulation using the MuJoCo physics engine, following the RoVerFly framework Kim et al. (2025). The quadrotor dynamics include motor lag, input delay, and randomized aerodynamic effects to reduce the sim-to-real gap. A RL policy is trained with task and domain randomization, enabling robust performance across varying payload masses and cable lengths without controller switching. Nominal parameters are summarized in Tab. 3.

**Failure Definition.** A rollout is marked as a crash if any of the following occurs: (i) ground contact ($z < 0.05$), (ii) Euler angles exceeding $60°$, or (iii) position error greater than 1 m. This definition captures both catastrophic instability and sustained loss of tracking in simulation.

## E RELATIONSHIP BETWEEN TRAJECTORY AND MODEL-LEVEL KL BUDGETS

In this section, we formalize the relationship between RAPO's two divergence budgets: the *trajectory-level* budget $\epsilon$ (controlling exponential tilting over next-states/trajectories under a fixed model) and the *model-level* budget $\kappa$ (controlling Boltzmann reweighting over an ensemble of dynamics models). We show both arise from a single robust optimization posed on the *joint* law over model indices and trajectories. By the chain rule of relative entropy, the joint KL cleanly decomposes into a model term and a (mean) trajectory term. This yields a principled trade-off between $\epsilon$ and $\kappa$, and justifies RAPO's two-temperature dual ($\eta$ for trajectory tilting, $\beta$ for model reweighting). We provide full statements and proofs.

### E.1 SETUP: JOINT NOMINAL LAW AND ADVERSARY

Let $\Omega$ be a measurable set of model indices with prior $\rho$ (discrete or continuous), and let $P_\omega$ denote the nominal trajectory law under model $\omega \in \Omega$ (e.g., a discounted path measure induced by the nominal transition kernel for that model). The joint nominal law on $(\omega, \tau)$ is $\hat{P}(\omega, \tau) = \rho(\omega)P_\omega(\tau)$. An adversary selects a joint law $Q$ on $(\omega, \tau)$, which we factor as $Q(\omega, \tau) = w(\omega)Q_\omega(\tau)$, where $w$ is a probability measure on $\Omega$ and $Q_\omega$ is a probability measure on trajectories for each $\omega$. We use the following performance functional on trajectories (e.g., a discounted return or robust value surrogate): $R(\tau) \in \mathbb{R}$, bounded with $R(\tau) \in [m, M]$. The robust objective (minimization of return or maximization of cost) will be expressed as an adversarial expectation of $R(\tau)$ under $Q$.

### E.2 A SINGLE JOINT KL BUDGET AND ITS EXACT DECOMPOSITION

**Definition 1 (Joint KL budget)** *Fix $B \geq 0$. The adversary is constrained by a single joint information budget $D_{\mathrm{KL}}(Q\|\hat{P}) \leq B$.*

**Lemma 1 (Chain rule decomposition of the joint KL)** *For any factorization $Q(\omega, \tau) = w(\omega)Q_\omega(\tau)$ and $\hat{P}(\omega, \tau) = \rho(\omega)P_\omega(\tau)$, $D_{\mathrm{KL}}(Q\|\hat{P}) = D_{\mathrm{KL}}(w\|\rho) + \mathbb{E}_{\omega \sim w}[D_{\mathrm{KL}}(Q_\omega\|P_\omega)]$.*

**Proof 1** *By definition, $D_{\mathrm{KL}}(Q\|\hat{P}) = \int Q(\omega, \tau) \log \frac{Q(\omega,\tau)}{\hat{P}(\omega,\tau)} \, \mathrm{d}\tau \, \mathrm{d}\omega$. Factoring and rearranging gives $D_{\mathrm{KL}}(Q\|\hat{P}) = D_{\mathrm{KL}}(w\|\rho) + \mathbb{E}_{\omega \sim w}[D_{\mathrm{KL}}(Q_\omega\|P_\omega)]$.*

**Corollary 1 (Additive budget split and worst-case bounds)** *Under the single-budget constraint $D_{\mathrm{KL}}(Q\|\hat{P}) \leq B$, any feasible $(w, \{Q_\omega\})$ satisfies $D_{\mathrm{KL}}(w\|\rho) + \mathbb{E}_{\omega \sim w}[D_{\mathrm{KL}}(Q_\omega\|P_\omega)] \leq B$. In particular, if one enforces $D_{\mathrm{KL}}(w\|\rho) \leq \kappa$ and $D_{\mathrm{KL}}(Q_\omega\|P_\omega) \leq \epsilon$ for $w$-almost every $\omega$, then $\kappa + \epsilon \leq B$ implies joint feasibility. Conversely, for any feasible $Q$, we must have $\kappa \leq B$ and $\mathbb{E}_w[D_{\mathrm{KL}}(Q_\omega\|P_\omega)] \leq B - \kappa$.*

**Interpretation.** Lemma shows that a single joint KL budget $B$ naturally decomposes into a model-level information shift $\kappa := D_{\mathrm{KL}}(w\|\rho)$ and an average trajectory-level information shift $\mathbb{E}_w[\cdot]$. Hence, RAPO's two budgets $(\kappa, \epsilon)$ can be viewed as a principled split of a single resource $B$.

### E.3 JOINT ROBUST OPTIMIZATION AND SEPARABLE DUAL

Consider the joint robust program $\inf_{Q: D_{\mathrm{KL}}(Q\|\hat{P}) \leq B} \mathbb{E}_{(\omega,\tau) \sim Q}[R(\tau)]$. By Lemma 1, introducing the factorization $Q = w \cdot Q_\omega$ and first-order Lagrange multipliers yields separability across *model* and *trajectory* levels.

**Theorem 1 (Dual of the joint program: two temperatures)** *Assume $R$ is bounded. The Lagrangian dual yields the equivalent unconstrained maximization over $(\beta, \eta) \geq 0$: $\inf_{Q: D_{\mathrm{KL}}(Q\|\hat{P}) \leq B} \mathbb{E}_Q[R] = \sup_{\beta,\eta \geq 0}\{-\frac{1}{\beta}\log\mathbb{E}_{\omega\sim\rho}[e^{-\beta h_\eta(\omega)}] - B/\beta\}$, where $h_\eta(\omega) := -(1/\eta)\log\mathbb{E}_{\tau\sim P_\omega}[e^{-\eta R(\tau)}]$ is the trajectory-level entropic risk. The maximizers $(\beta^\star, \eta^\star)$ induce Boltzmann reweighting $w_{\beta^\star}(\omega) \propto \rho(\omega)e^{-\beta^\star h_{\eta^\star}(\omega)}$ and exponential tilting $Q_\omega^{\eta^\star}(d\tau) \propto e^{-\eta^\star R(\tau)}P_\omega(d\tau)$, with complementary-slackness condition $D_{\mathrm{KL}}(w_{\beta^\star}\|\rho) + \mathbb{E}_{\omega\sim w_{\beta^\star}}[D_{\mathrm{KL}}(Q_\omega^{\eta^\star}\|P_\omega)] = B$.*

**Proof 2** *Form the Lagrangian with multiplier $\lambda \geq 0$: $\mathcal{L}(Q, \lambda) = \int Q(\omega, \tau)R(\tau)\,d\omega d\tau + \lambda(\int Q\log\frac{Q}{\hat{P}} - B)$. Minimization over $Q$ gives $Q^\lambda(\omega, \tau) \propto \hat{P}(\omega, \tau)e^{-R(\tau)/\lambda} = \rho(\omega)P_\omega(\tau)e^{-R(\tau)/\lambda}$. Writing $\eta = \lambda^{-1}$ defines $h_\eta(\omega) = -(1/\eta)\log\mathbb{E}_{P_\omega}[e^{-\eta R}]$. The remaining dependence on $\omega$ yields $-(1/\beta)\log\mathbb{E}_\rho[e^{-\beta h_\eta(\omega)}]$ with $\beta = \lambda$. Strong duality holds by convexity and Slater's condition. Optimality yields the reported Boltzmann/tilting forms and tightness of the joint KL.*

**Mapping to RAPO.** A single joint constraint gives rise to two dual temperatures: $\eta \leftrightarrow$ trajectory tilting (soft minimum within a model), $\beta \leftrightarrow$ model reweighting (soft minimum across models).

Identifying $D_{\mathrm{KL}}(w\|\rho) = \kappa$ and $\mathbb{E}_w[D_{\mathrm{KL}}(Q_\omega\|P_\omega)]$ as the trajectory budget (with worst-case upper bound $\epsilon$) gives the relation $\kappa + \epsilon \leq B$.

### E.4 Trade-offs, Bounds, and Practical Allocations

**Proposition 1 (Budget trade-off)** *Let $B$ be fixed. If $D_{\mathrm{KL}}(w\|\rho) \leq \kappa$ and $D_{\mathrm{KL}}(Q_\omega\|P_\omega) \leq \epsilon$ for all $\omega$, then $\kappa + \epsilon \leq B$. Conversely, any feasible $(w, \{Q_\omega\})$ satisfies $\kappa \leq B$ and $\mathbb{E}_w[\text{traj-KL}] \leq B - \kappa$.*

**Proof 3** *Immediate from Lemma 1 and Corollary 1.*

**Design guidance.**   With a single budget $B$, two allocation rules emerge: (i) Fixed split $\kappa = \alpha B$, $\epsilon = (1 - \alpha)B$ for $\alpha \in [0, 1]$, reflecting prior trust. (ii) Adaptive split via dual updates: $\beta \leftarrow \beta + \gamma_\beta(D_{\mathrm{KL}}(w\|\rho) - \kappa)$, $\eta \leftarrow \eta + \gamma_\eta(\mathbb{E}_w[\text{traj-KL}] - \epsilon)$, i.e., Robbins–Monro updates compatible with RAPO.

### E.5 Specializations: Hard vs. Soft, Per-Step vs. Trajectory

**Hard vs. soft.**   The hard joint constraint $D_{\mathrm{KL}}(Q\|\hat{P}) \leq B$ yields the explicit $-B/\beta$ term and implicit trajectory free energy $h_\eta(\omega)$. Replacing hard constraints by penalties removes explicit offsets but preserves the two-temperature structure. KL residuals then appear as regularizers in RAPO.

**Trajectory vs. per-step KL.**   If $Q_\omega, P_\omega$ arise from Markov kernels, the trajectory KL decomposes as $D_{\mathrm{KL}}(Q_\omega\|P_\omega) = \mathbb{E}_{Q_\omega}[\sum_{t\geq 0} \gamma^t D_{\mathrm{KL}}(q_\omega(\cdot|s_t, a_t)\|p_\omega(\cdot|s_t, a_t))]$. Thus uniform one-step budgets $\delta_t \leq \bar{\delta}$ imply $\epsilon \leq \sum_{t\geq 0} \gamma^t \bar{\delta} = \bar{\delta}/(1 - \gamma)$, mapping per-step to trajectory-level budgets.

### E.6 Summary

A single joint KL budget decomposes into a model-level divergence and a mean trajectory-level divergence. The dual factorizes into two entropic risks with dual temperatures $(\beta, \eta)$, matching RAPO's Boltzmann reweighting across models and exponential tilting within each model. This yields $\kappa + \epsilon \leq B$ (or $\kappa + \mathbb{E}_w[\text{traj-KL}] \leq B$) and motivates fixed or adaptive allocation strategies, justifying the two-budget design from a unified robust optimization principle.

## F  Finite Ensemble Approximation of Model Uncertainty

RAPO models dynamics uncertainty at the model level by a *finite ensemble* of $K$ dynamics models sampled from a nominal prior. Theoretically, uncertainty lives on a continuous parameter space $(\Omega, \rho)$, and the ensemble is an approximation. This section establishes that the finite-ensemble robust operator and its optimizer converge to their continuous-space counterparts as $K \to \infty$, with rates matching standard Monte Carlo concentration.

### F.1 Setup and Normalization

Let $\Omega$ be a measurable parameter space with prior $\rho$. For each $\omega \in \Omega$, let $h(\omega)$ denote the (one-step) robust "energy" we tilt with. We assume:

**Assumption 1 (Bounded range and nondegenerate variance)** *There exist $m < M$ such that $h(\omega) \in [m, M]$ for all $\omega$. Define $H := M - m$. Without loss of generality we re-center $h$ by $h \leftarrow h - m$, so $h \in [0, H]$. In addition, there exists $\sigma^2 > 0$ and a compact interval $\mathcal{B} = [\beta_{\min}, B] \subset (0, \infty)$ such that for all $\beta \in \mathcal{B}$, the tilted law $w_\beta(d\omega) \propto e^{-\beta h(\omega)}\rho(d\omega)$ satisfies $\mathrm{Var}_{w_\beta}(h) \geq \sigma^2$.*

For $\beta \geq 0$, define $Z(\beta) = \mathbb{E}_\rho[e^{-\beta h(\omega)}]$, $\Phi(\beta) = -\frac{1}{\beta}\log Z(\beta) - \frac{\kappa}{\beta}$, with continuous tilted distribution $w_\beta(d\omega) = \frac{e^{-\beta h(\omega)}}{Z(\beta)}\rho(d\omega)$.

Given an i.i.d. ensemble $\{\omega_k\}_{k=1}^K \sim \rho$, let $\rho_K = \frac{1}{K} \sum_{k=1}^K \delta_{\omega_k}$. The finite-ensemble analogues are $Z_K(\beta) = \frac{1}{K} \sum_{k=1}^K e^{-\beta h(\omega_k)}$, $\Phi_K(\beta) = -\frac{1}{\beta} \log Z_K(\beta) - \frac{\kappa}{\beta}$, and discrete tilted weights $w_{\beta,K}(k) = \frac{e^{-\beta h(\omega_k)}}{\sum_{j=1}^K e^{-\beta h(\omega_j)}}$.

**Basic envelopes and Lipschitz constants.** Under Assumption 1, $e^{-BH} \leq e^{-\beta h(\omega)} \leq 1$, and $|\partial_\beta e^{-\beta h(\omega)}| = h(\omega) e^{-\beta h(\omega)} \leq H$. Hence $f_\beta(\omega) := e^{-\beta h(\omega)}$ is bounded by 1 and $H$-Lipschitz in $\beta$ on $\mathcal{B}$.

## F.2 Uniform Concentration of the Log-Partition

**Theorem 2 (Uniform concentration of $Z_K$ and $\log Z_K$)** *Under Assumption 1, for any $\delta \in (0,1)$, with probability at least $1 - \delta$, $\sup_{\beta \in \mathcal{B}} |Z_K(\beta) - Z(\beta)| \leq C_1 \sqrt{\frac{\log(2N/\delta)}{K}} + H\Delta$, where $N := \lceil (B - \beta_{\min})/\Delta \rceil$, $\Delta > 0$ is arbitrary, and $C_1 := 1$. Choosing $\Delta = \sqrt{\frac{\log(2N/\delta)}{K}}$ yields $\sup_{\beta \in \mathcal{B}} |Z_K(\beta) - Z(\beta)| = O(\sqrt{\frac{\log(1/\delta)}{K}})$. Moreover, since $Z(\beta), Z_K(\beta) \in [e^{-BH}, 1]$, we also have $\sup_{\beta \in \mathcal{B}} |\log Z_K(\beta) - \log Z(\beta)| \leq e^{BH} \sup_{\beta \in \mathcal{B}} |Z_K(\beta) - Z(\beta)| = O(\sqrt{\frac{\log(1/\delta)}{K}})$.*

**Proof 4** *Fix $\Delta > 0$ and construct a grid $\{\beta_j\}_{j=1}^N$ on $\mathcal{B}$ with mesh $\leq \Delta$. For each $\beta_j$, Hoeffding's inequality gives $\mathbb{P}(|Z_K(\beta_j) - Z(\beta_j)| > \epsilon) \leq 2e^{-2K\epsilon^2}$. A union bound over $N$ points yields $\mathbb{P}(\max_j |Z_K(\beta_j) - Z(\beta_j)| > \epsilon) \leq 2Ne^{-2K\epsilon^2}$. Set $\epsilon = \sqrt{\frac{\log(2N/\delta)}{2K}}$. For any $\beta$, pick nearest $\beta_j$; Lipschitzness gives $|Z_K(\beta) - Z(\beta)| \leq H\Delta + \epsilon + H\Delta$. Absorb constants to obtain the bound with $C_1 = 1$. For $\log Z$, use $|\log x - \log y| \leq |x - y|/\min\{x, y\}$ and $\min\{Z(\beta), Z_K(\beta)\} \geq e^{-BH}$.*

**Corollary 2 (Uniform concentration of $\Phi_K$)** *Under Assumption 1, with prob. $\geq 1 - \delta$, $\sup_{\beta \in \mathcal{B}} |\Phi_K(\beta) - \Phi(\beta)| \leq \frac{1}{\beta_{\min}} \sup_{\beta \in \mathcal{B}} |\log Z_K(\beta) - \log Z(\beta)| = O(\sqrt{\frac{\log(1/\delta)}{K}})$.*

**Proof 5** *Immediate from $\Phi(\beta) = -\frac{1}{\beta} \log Z(\beta) - \frac{\kappa}{\beta}$ and $\beta \geq \beta_{\min}$.*

## F.3 Stability of the Optimal Dual Parameter

Let $\beta^\star = \arg\max_{\beta \in \mathcal{B}} \Phi(\beta)$ and $\beta_K^\star = \arg\max_{\beta \in \mathcal{B}} \Phi_K(\beta)$ (existence follows from compactness). We first recall strong concavity of $\Phi$.

**Lemma 2 (Strong concavity of $\Phi$)** *Under Assumption 1, for all $\beta \in \mathcal{B}$, $\Phi''(\beta) = -\frac{1}{\beta} \text{Var}_{w_\beta}(h) - \frac{2}{\beta^3}(\kappa - D_{\text{KL}}(w_\beta \| \rho)) \leq -\frac{\sigma^2}{B} =: -\mu$. In particular, $\Phi$ is $\mu$-strongly concave on $\mathcal{B}$.*

**Proof 6** *Let $A(\beta) := \log Z(\beta)$. Standard exponential-family identities give $A'(\beta) = -\mathbb{E}_{w_\beta}[h]$ and $A''(\beta) = \text{Var}_{w_\beta}(h)$. Since $\Phi(\beta) = -(1/\beta)A(\beta) - (\kappa/\beta)$, straightforward differentiation yields $\Phi'(\beta) = \frac{A(\beta) + \kappa}{\beta^2} + \frac{1}{\beta} \mathbb{E}_{w_\beta}[h] = \frac{\kappa + \log Z(\beta) + \beta \mathbb{E}_{w_\beta}[h]}{\beta^2} = \frac{\kappa - D_{\text{KL}}(w_\beta \| \rho)}{\beta^2}$, using $D_{\text{KL}}(w_\beta \| \rho) = -\log Z(\beta) - \beta \mathbb{E}_{w_\beta}[h]$. Differentiating once more and using $D'_{\text{KL}}(w_\beta \| \rho) = \beta \text{Var}_{w_\beta}(h)$ gives $\Phi''(\beta) = -\frac{2}{\beta^3}(\kappa - D_{\text{KL}}(w_\beta \| \rho)) - \frac{1}{\beta} \text{Var}_{w_\beta}(h) \leq -\frac{1}{\beta} \text{Var}_{w_\beta}(h) \leq -\frac{\sigma^2}{B}$.*

**Theorem 3 (Argmax stability)** *Under Assumption 1, there exists a universal constant $C > 0$ (depending only on $\beta_{\min}, B, H, \sigma$) such that, with probability $1 - \delta$, $|\beta_K^\star - \beta^\star| \leq C\sqrt{\frac{\log(1/\delta)}{K}}$, and $|\Phi(\beta_K^\star) - \Phi(\beta^\star)| \leq \frac{\mu}{2} |\beta_K^\star - \beta^\star|^2 = O(\frac{\log(1/\delta)}{K})$.*

**Proof 7** *By Corollary 2, $\sup_\beta |\Phi_K - \Phi| \leq \varepsilon_K$ with $\varepsilon_K = O(\sqrt{\frac{\log(1/\delta)}{K}})$. By strong concavity (Lemma 2), for any $x \in \mathcal{B}$, $\Phi(\beta^\star) - \Phi(x) \geq \frac{\mu}{2} |\beta^\star - x|^2$. Since $\Phi_K(\beta_K^\star) \geq \Phi_K(\beta^\star)$, $\Phi(\beta^\star) - \Phi(\beta_K^\star) \leq |\Phi(\beta^\star) - \Phi_K(\beta^\star)| + |\Phi_K(\beta_K^\star) - \Phi(\beta_K^\star)| \leq 2\varepsilon_K$. Combining, $\frac{\mu}{2} |\beta_K^\star - \beta^\star|^2 \leq \Phi(\beta^\star) -$

$\Phi(\beta_K^\star) \le 2\varepsilon_K$, *hence* $|\beta_K^\star - \beta^\star| \le \sqrt{4\varepsilon_K/\mu}$. *The value gap bound follows by strong concavity:* $\Phi(\beta^\star) - \Phi(\beta_K^\star) \le \frac{\mu}{2}|\beta^\star - \beta_K^\star|^2$.

## F.4 Convergence of Tilted Expectations (Weak Convergence)

While $w_{\beta,K}$ is supported on the finite set $\{\omega_k\}$, $w_\beta$ is continuous on $\Omega$. Comparing them in total variation is not meaningful. The correct notion is convergence of expectations for bounded test functions.

**Theorem 4 (Uniform convergence of tilted expectations)** *Let $\mathcal{G}$ be any class of bounded measurable test functions $g : \Omega \to \mathbb{R}$ with $\|g\|_\infty \le 1$. Under Assumption 1, for any fixed $\beta \in \mathcal{B}$, $\sup_{g \in \mathcal{G}} |\mathbb{E}_{w_{\beta,K}}[g(\omega)] - \mathbb{E}_{w_\beta}[g(\omega)]| = O_p(K^{-1/2})$. Consequently, replacing $\beta$ by $\beta_K^\star$ and $\beta^\star$ preserves the $O_p(K^{-1/2})$ rate by Theorem 3.*

**Proof 8** *For fixed $\beta$, write the tilted expectations as ratios:* $\mathbb{E}_{w_{\beta,K}}[g] = \frac{\frac{1}{K}\sum_k e^{-\beta h(\omega_k)} g(\omega_k)}{Z_K(\beta)}$, $\mathbb{E}_{w_\beta}[g] = \frac{\mathbb{E}_\rho[e^{-\beta h} g]}{Z(\beta)}$. *Both numerator and denominator are empirical means of bounded variables (envelopes $\le 1$). By the same grid/union bound argument as in Theorem ?? (now without grid since $\beta$ is fixed), each concentrates at rate $O_p(K^{-1/2})$. A standard ratio bound yields* $|\mathbb{E}_{w_{\beta,K}}[g] - \mathbb{E}_{w_\beta}[g]| \le \frac{|\frac{1}{K}\sum e^{-\beta h} g - \mathbb{E}[e^{-\beta h} g]|}{Z(\beta)} + \frac{\mathbb{E}[e^{-\beta h}|g|]}{Z(\beta)Z_K(\beta)}|Z_K(\beta) - Z(\beta)|$, *and* $Z(\beta), Z_K(\beta) \ge e^{-BH}$ *lower bound the denominators. Both terms are $O_p(K^{-1/2})$ uniformly over $\|g\|_\infty \le 1$, completing the proof.*

## F.5 Operator- and Value-Level Convergence

Let $\Psi_\beta(V) := -(1/\beta)\log \mathbb{E}_\rho[e^{-\beta V}]$ and its finite-ensemble version $\Psi_{\beta,K}(V) := -(1/\beta)\log \mathbb{E}_{\rho_K}[e^{-\beta V}]$. Fix discount $\gamma \in (0,1)$. Consider the (model-level) hard-budget robust operators on bounded value functions: $(\mathcal{T}V)(s) = \max_a \sup_{\beta \in \mathcal{B}}\{ r(s,a) + \gamma(\Psi_\beta(V \mid s,a) - \kappa/\beta) \}$, $(\mathcal{T}_K V)(s) = \max_a \sup_{\beta \in \mathcal{B}}\{ r(s,a) + \gamma(\Psi_{\beta,K}(V \mid s,a) - \kappa/\beta) \}$. (Exactly the same arguments apply to the soft-penalty form, replacing $-\kappa/\beta$ with 0.)

**Theorem 5 (Uniform operator convergence)** *Under Assumption 1, for any bounded $V$ with range contained in $[0, H_V]$, there exists $C > 0$ such that, with probability at least $1 - \delta$, $\|\mathcal{T}_K V - \mathcal{T}V\|_\infty \le \frac{\gamma}{\beta_{\min}}\sup_{(s,a),\,\beta \in \mathcal{B}}|\log Z_{V,K}(\beta; s,a) - \log Z_V(\beta; s,a)| \le C\gamma\sqrt{\frac{\log(1/\delta)}{K}}$, where $Z_V(\beta; s,a) = \mathbb{E}_\rho[e^{-\beta V(s')}]$ and $Z_{V,K}$ is its empirical analogue.*

**Proof 9** *For any $(s,a)$, the maps $\beta \mapsto \Psi_\beta(V \mid s,a) - \kappa/\beta$ and its empirical counterpart are uniformly approximated at rate $O(\sqrt{\frac{\log(1/\delta)}{K}})$ by Theorem 5 (applied to $h \equiv V(s') \in [0, H_V]$). Taking $\sup_\beta$ preserves the deviation bound: $\sup_{\beta \in \mathcal{B}}|\Psi_{\beta,K}(V \mid s,a) - \Psi_\beta(V \mid s,a)| \le \frac{1}{\beta_{\min}}\sup_{\beta \in \mathcal{B}}|\log Z_{V,K}(\beta) - \log Z_V(\beta)|$. The outer $\max_a$ also preserves the bound, and the reward terms cancel. Uniformity in $(s,a)$ follows from the common envelope bounds for $V$.*

**Corollary 3 (Value fixed-point convergence)** *Let $V^\star$ and $V_K^\star$ be the unique fixed points of $\mathcal{T}$ and $\mathcal{T}_K$ (both $\gamma$-contractions under the $\|\cdot\|_\infty$ norm). Then, with probability at least $1 - \delta$, $\|V_K^\star - V^\star\|_\infty \le \frac{1}{1-\gamma}\|\mathcal{T}_K V^\star - \mathcal{T}V^\star\|_\infty = O(\sqrt{\frac{\log(1/\delta)}{K}})$.*

**Proof 10** *By the contraction mapping perturbation bound, $\|V_K^\star - V^\star\|_\infty \le \frac{1}{1-\gamma}\sup_V \|\mathcal{T}_K V - \mathcal{T}V\|_\infty$, and invoke Theorem 5.*

## F.6 Policy-Level Consequences

Let $\pi^\star$ and $\pi_K^\star$ be greedy w.r.t. $V^\star$ and $V_K^\star$ under the robust operators above (or any standard tie-breaking). Using standard performance difference arguments adapted to the robust setting (omitted

for brevity), one obtains for the robust return $J(\pi)$: $|J(\pi_K^\star) - J(\pi^\star)| \leq \frac{2\gamma}{1-\gamma}\|V_K^\star - V^\star\|_\infty = O(\sqrt{1/K})$. (The proof follows the classical performance difference lemma (PDL) with the robust Bellman residual and the contraction and bounded range ensure the same constants up to $\gamma$.)

### F.7 DISCUSSION AND PRACTICAL IMPLICATIONS

The results above show that RAPO's finite-ensemble approximation is theoretically sound. As $K$ grows, the finite-ensemble dual $\Phi_K$ uniformly approximates $\Phi$. The optimizer $\beta_K^\star$ converges to $\beta^\star$ at the Monte Carlo rate. Tilted expectations under $w_{\beta,K}$ converge to those under $w_\beta$ for any bounded test function. The robust Bellman operator, its fixed point, and the induced policy converge accordingly. The constants depend on the value range, the $\beta$-window $[\beta_{\min}, B]$, and a variance lower bound $\sigma^2$ which encodes non-degeneracy of the robust tilting. In practice, moderate ensemble sizes (e.g., $K \in [8, 32]$) already yield stable gains and our empirical sensitivity in Tab. 2 is consistent with the $O(K^{-1/2})$ scaling predicted here.

## G PROPERTIES OF ROBUST BELLMAN OPERATOR

### G.1 OPERATOR PROPERTIES

We begin by establishing fundamental algebraic properties of the robust Bellman operator. These properties ensure that the operator retains the same structural behavior as its nominal counterpart, thereby enabling dynamic programming arguments in the robust setting.

Recall the robust Bellman operator for a fixed policy $\pi$:

$$(\mathcal{T}_\mathcal{P}^\pi V)(s) = \mathbb{E}_{a\sim\pi(\cdot|s)}\left[r(s,a) + \gamma \inf_{p(\cdot|s,a)\in\mathcal{P}_\epsilon(s,a)} \mathbb{E}_{s'\sim p(\cdot|s,a)}V(s')\right]. \tag{11}$$

**Lemma 3 (Monotonicity)** *For any two bounded functions $V, W : \mathcal{S} \to \mathbb{R}$, if $V(s) \leq W(s)$ for all $s \in \mathcal{S}$, then $(\mathcal{T}_\mathcal{P}^\pi V)(s) \leq (\mathcal{T}_\mathcal{P}^\pi W)(s)$ for all $s \in \mathcal{S}$.*

**Proof 11** *Fix $s \in \mathcal{S}$. For each $a \in \mathcal{A}$, consider the inner infimum $\inf_{p\in\mathcal{P}_\epsilon(s,a)} \mathbb{E}_{s'\sim p}[V(s')]$. Since $V \leq W$ pointwise, we have $\mathbb{E}_p[V(s')] \leq \mathbb{E}_p[W(s')]$ for any distribution $p$. Taking the infimum over the same set preserves the inequality. Averaging over $a \sim \pi(\cdot|s)$ and adding $r(s,a)$ and $\gamma$ yields the claim.*

**Lemma 4 (Translation Invariance)** *For any bounded $V : \mathcal{S} \to \mathbb{R}$ and constant $c \in \mathbb{R}$,*

$$\mathcal{T}_\mathcal{P}^\pi(V + c)(s) = \mathcal{T}_\mathcal{P}^\pi V(s) + \gamma c, \quad \forall s \in \mathcal{S}. \tag{12}$$

**Proof 12** *Fix $s, a$. For any $p \in \mathcal{P}_\epsilon(s,a)$, $\mathbb{E}_{s'\sim p}[V(s') + c] = \mathbb{E}_{s'\sim p}[V(s')] + c$. Taking the infimum over $p$ preserves equality. Plugging back into the definition of $\mathcal{T}_\mathcal{P}^\pi$ and multiplying the additive constant by $\gamma$ yields the claim.*

**Proposition 2 ($\gamma$-Contraction)** *For any bounded $V, W : \mathcal{S} \to \mathbb{R}$,*

$$\|\mathcal{T}_\mathcal{P}^\pi V - \mathcal{T}_\mathcal{P}^\pi W\|_\infty \leq \gamma\|V - W\|_\infty. \tag{13}$$

**Proof 13** *Fix $s \in \mathcal{S}$. For any $a \in \mathcal{A}$ and any $p \in \mathcal{P}_\epsilon(s,a)$, $|\mathbb{E}_{s'\sim p}[V(s')] - \mathbb{E}_{s'\sim p}[W(s')]| \leq \|V - W\|_\infty$. Taking the infimum over $p$ preserves the inequality. Therefore, for each $s$, $|(\mathcal{T}_\mathcal{P}^\pi V)(s) - (\mathcal{T}_\mathcal{P}^\pi W)(s)| \leq \gamma\|V - W\|_\infty$. Taking the supremum over $s$ yields the claim.*

**Corollary 4 (Existence and uniqueness of robust value)** *By Banach's fixed-point theorem, $\mathcal{T}_\mathcal{P}^\pi$ admits a unique fixed point $V_{\mathcal{P}_\epsilon}^\pi$ in $\mathbb{R}^\mathcal{S}$. Moreover, value iteration $V_{k+1} = \mathcal{T}_\mathcal{P}^\pi V_k$ converges to $V_{\mathcal{P}_\epsilon}^\pi$ for any initialization $V_0$.*

**Discussion.** The above properties mirror those of the nominal Bellman operator, with the only difference being the inner minimization over the uncertainty set $\mathcal{P}_\epsilon(s,a)$. These results ensure that standard dynamic programming arguments extend naturally to the robust setting, thereby justifying the use of value iteration and policy iteration under distributional ambiguity.

## G.2 STATIONARY OPTIMALITY AND WORST-CASE KERNELS

We now establish that in the robust MDP setting, it suffices to optimize over stationary deterministic policies. Moreover, for any fixed stationary policy, there exists a stationary worst-case kernel that attains the inner minimization in the robust Bellman operator.

**Proposition 3 (Existence of stationary robust optimal policies)** *Consider a robust MDP with state space $\mathcal{S}$, action space $\mathcal{A}$, bounded reward function $r : \mathcal{S} \times \mathcal{A} \to [0, 1]$, and discount factor $\gamma \in (0, 1)$. Suppose that for each $(s, a) \in \mathcal{S} \times \mathcal{A}$ the uncertainty set $\mathcal{P}_\epsilon(s, a)$ is nonempty, compact, and convex, and that rectangularity holds across $(s, a)$. Then there exists a stationary deterministic policy $\pi^\star$ such that*

$$V_{\mathcal{P}_\epsilon}^{\pi^\star}(s) = \sup_\pi V_{\mathcal{P}_\epsilon}^{\pi}(s), \quad \forall s \in \mathcal{S}. \tag{14}$$

**Proof 14** *Define the robust optimality operator*

$$(\mathcal{T}_{\mathcal{P}} V)(s) = \sup_{a \in \mathcal{A}} \left\{ r(s, a) + \gamma \inf_{p \in \mathcal{P}_\epsilon(s,a)} \mathbb{E}_{s' \sim p}[V(s')] \right\}. \tag{15}$$

*We first note that the operator $\mathcal{T}_{\mathcal{P}}$ is a $\gamma$-contraction on the space of bounded functions (Proposition 2), hence it admits a unique fixed point $V^\star$. Moreover, by Banach's theorem, value iteration under $\mathcal{T}_{\mathcal{P}}$ converges to $V^\star$ from any initialization.*

*Next, fix a state $s \in \mathcal{S}$. For each action $a$, the inner minimization problem*

$$\inf_{p \in \mathcal{P}_\epsilon(s,a)} \mathbb{E}_{s' \sim p}[V(s')] \tag{16}$$

*is a continuous convex optimization over the compact convex set $\mathcal{P}_\epsilon(s, a)$. By the Weierstrass theorem, the minimum is attained by some distribution $p^\star(\cdot|s, a)$. Thus, for each $a$, the inner infimum is well defined.*

*Now, consider the outer maximization over actions. The map*

$$a \mapsto r(s, a) + \gamma \inf_{p \in \mathcal{P}_\epsilon(s,a)} \mathbb{E}_{s' \sim p}[V(s')] \tag{17}$$

*is linear in the choice of a distribution over $\mathcal{A}$. Hence, the supremum over randomized policies $\pi(\cdot|s)$ is attained at an extreme point of the simplex over $\mathcal{A}$, i.e., a deterministic action. Therefore, for each state $s$, the optimal choice is deterministic.*

*It follows that dynamic programming can be restricted to deterministic stationary policies. The greedy policy $\pi^\star$ defined by*

$$\pi^\star(s) = \arg\max_{a \in \mathcal{A}} \left\{ r(s, a) + \gamma \inf_{p \in \mathcal{P}_\epsilon(s,a)} \mathbb{E}_{s' \sim p}[V^\star(s')] \right\} \tag{18}$$

*is stationary and deterministic, and it achieves $V^\star$. Thus $\pi^\star$ is a robust optimal policy.*

**Proposition 4 (Existence of stationary worst-case kernels)** *Fix any stationary policy $\pi$. Under the same assumptions as Proposition 3, there exists a stationary kernel $p_\pi^{\mathrm{wc}} \in \mathcal{P}_\epsilon$ such that*

$$V_{\mathcal{P}_\epsilon}^{\pi}(s) = V_{p_\pi^{\mathrm{wc}}}^{\pi}(s), \quad \forall s \in \mathcal{S}. \tag{19}$$

**Proof 15** *Consider the robust evaluation operator*

$$(\mathcal{T}_{\mathcal{P}}^{\pi} V)(s) = \mathbb{E}_{a \sim \pi(\cdot|s)} \left[ r(s, a) + \gamma \inf_{p \in \mathcal{P}_\epsilon(s,a)} \mathbb{E}_{s' \sim p}[V(s')] \right]. \tag{20}$$

*For each $(s, a)$, the set $\mathcal{P}_\epsilon(s, a)$ is nonempty, compact, and convex. The function $p \mapsto \mathbb{E}_{s' \sim p}[V(s')]$ is continuous and linear in $p$. By the extreme value theorem, the infimum is attained by some $p^\star(\cdot|s, a) \in \mathcal{P}_\epsilon(s, a)$. Define a kernel $p_\pi^{\mathrm{wc}}$ by*

$$p_\pi^{\mathrm{wc}}(\cdot|s, a) = p^\star(\cdot|s, a), \quad \forall(s, a). \tag{21}$$

*This kernel is stationary, since it depends only on the current state-action pair.*

*Now, evaluate the value function of $\pi$ under kernel $p_\pi^{\mathrm{wc}}$:*

$$V_{p_\pi^{\mathrm{wc}}}^\pi(s) = \mathbb{E}_{a\sim\pi(\cdot|s)}\Big[r(s,a) + \gamma\mathbb{E}_{s'\sim p_\pi^{\mathrm{wc}}(\cdot|s,a)}V_{p_\pi^{\mathrm{wc}}}^\pi(s')\Big]. \tag{22}$$

*By construction of $p_\pi^{\mathrm{wc}}$, the inner expectation coincides with the infimum defining $\mathcal{T}_{\mathcal{P}}^\pi V$. Hence,*

$$(\mathcal{T}_{\mathcal{P}}^\pi V)(s) = \mathbb{E}_{a\sim\pi(\cdot|s)}\Big[r(s,a) + \gamma\mathbb{E}_{s'\sim p_\pi^{\mathrm{wc}}(\cdot|s,a)}V(s')\Big]. \tag{23}$$

*Thus, the fixed point of $\mathcal{T}_{\mathcal{P}}^\pi$ coincides with the fixed point of the standard Bellman operator under $p_\pi^{\mathrm{wc}}$. Therefore,*

$$V_{\mathcal{P}_\epsilon}^\pi(s) = V_{p_\pi^{\mathrm{wc}}}^\pi(s), \quad \forall s \in \mathcal{S}. \tag{24}$$

**Discussion.** These results confirm that the robust control problem admits a well-defined saddle-point structure: the outer maximization over policies is achieved by a stationary deterministic policy, while the inner minimization over models is achieved by a stationary worst-case kernel. This structure justifies the design of RAPO and similar algorithms as policy iteration schemes under rectangular KL-uncertainty.

### G.3 ROBUST PERFORMANCE DIFFERENCE LEMMA

We now present the analogue of the PDL in the robust MDP setting. The result shows that the difference in robust returns between two policies can be expressed in terms of robust advantages weighted by the robust occupancy measure of the new policy Kakade & Langford (2002).

**Robust discounted occupancy measure.** For a stationary policy $\pi$ and its corresponding stationary worst-case kernel $p_\pi^{\mathrm{wc}}$ (Proposition 4), define the robust discounted state occupancy measure

$$d_{\mathcal{P}_\epsilon}^\pi(s) = (1-\gamma)\sum_{t=0}^\infty \gamma^t\,\mathbb{P}_{p_\pi^{\mathrm{wc}},\pi}(s_t = s). \tag{25}$$

This is a valid distribution over $\mathcal{S}$ and places higher weight on states visited earlier under $(\pi, p_\pi^{\mathrm{wc}})$.

**Robust advantage.** Given $\pi$ and $p_\pi^{\mathrm{wc}}$, define the robust action-value and advantage functions by

$$Q_{\mathcal{P}_\epsilon}^\pi(s,a) := r(s,a) + \gamma\,\mathbb{E}_{s'\sim p_\pi^{\mathrm{wc}}(\cdot|s,a)}[V_{\mathcal{P}_\epsilon}^\pi(s')], \tag{26}$$

and $A_{\mathcal{P}_\epsilon}^\pi(s,a) := Q_{\mathcal{P}_\epsilon}^\pi(s,a) - V_{\mathcal{P}_\epsilon}^\pi(s)$.

**Lemma 5 (Robust Performance Difference Lemma (RPDL))** *For any two stationary policies $\pi$ and $\pi'$, let $p_\pi^{\mathrm{wc}}$ and $p_{\pi'}^{\mathrm{wc}}$ denote their respective stationary worst-case kernels. Then*

$$J(\pi', p_{\pi'}^{\mathrm{wc}}) - J(\pi, p_\pi^{\mathrm{wc}}) = \frac{1}{1-\gamma}\,\mathbb{E}_{s\sim d_{\mathcal{P}_\epsilon}^{\pi'},\,a\sim\pi'(\cdot|s)}\big[A_{\mathcal{P}_\epsilon}^\pi(s,a)\big]. \tag{27}$$

**Proof 16** *We expand the robust return of $\pi'$:*

$$J(\pi', p_{\pi'}^{\mathrm{wc}}) = \mathbb{E}_{s_0\sim\mu_0}[V_{\mathcal{P}_\epsilon}^{\pi'}(s_0)], \tag{28}$$

*where $\mu_0$ is the initial distribution. For each state $s$,*

$$V_{\mathcal{P}_\epsilon}^\pi(s) = \mathbb{E}_{a\sim\pi(\cdot|s)}\Big[r(s,a) + \gamma\,\mathbb{E}_{s'\sim p_\pi^{\mathrm{wc}}(\cdot|s,a)}V_{\mathcal{P}_\epsilon}^\pi(s')\Big]. \tag{29}$$

*Define the robust temporal-difference error of $\pi$ evaluated under $\pi'$ and $p_{\pi'}^{\mathrm{wc}}$:*

$$\delta^\pi(s,a,s') := r(s,a) + \gamma V_{\mathcal{P}_\epsilon}^\pi(s') - V_{\mathcal{P}_\epsilon}^\pi(s). \tag{30}$$

*By construction, $\mathbb{E}_{s'\sim p_\pi^{\mathrm{wc}}(\cdot|s,a)}[\delta^\pi(s,a,s')] = A_{\mathcal{P}_\epsilon}^\pi(s,a)$.*

*Now unroll the recursion for $V_{\mathcal{P}_\epsilon}^{\pi'}$ along a trajectory $(s_t,a_t)$ generated by $(\pi', p_{\pi'}^{\mathrm{wc}})$:*

$$V_{\mathcal{P}_\epsilon}^{\pi'}(s_0) - V_{\mathcal{P}_\epsilon}^\pi(s_0) = \mathbb{E}\Big[\sum_{t=0}^\infty \gamma^t\big(r(s_t,a_t) + \gamma V_{\mathcal{P}_\epsilon}^\pi(s_{t+1}) - V_{\mathcal{P}_\epsilon}^\pi(s_t)\big)\,\Big|\,s_0\Big] \tag{31}$$

$$= \mathbb{E}\Big[\sum_{t=0}^\infty \gamma^t\,\delta^\pi(s_t,a_t,s_{t+1})\,\Big|\,s_0\Big]. \tag{32}$$

*Taking expectations over $s_0 \sim \mu_0$ and re-indexing with the definition of $d_{\mathcal{P}_\epsilon}^{\pi'}$, we obtain*

$$J(\pi', p_{\pi'}^{\mathrm{wc}}) - J(\pi, p_\pi^{\mathrm{wc}}) = \frac{1}{1-\gamma} \mathbb{E}_{s \sim d_{\mathcal{P}_\epsilon}^{\pi'}, \, a \sim \pi'(\cdot|s)}[A_{\mathcal{P}_\epsilon}^\pi(s,a)]. \tag{33}$$

**Discussion.**   The lemma reveals a saddle-point structure: the robust performance improvement of $\pi'$ over $\pi$ depends only on the robust advantages of $\pi$, evaluated under the robust occupancy measure induced by $\pi'$. This extends the classical PDL to the distributionally robust setting, justifying robust policy improvement schemes where policy updates are guided by robust advantage estimates.

### G.4   ROBUST VALUE DROP BOUND

We now quantify how much the robust value under the KL-ball $\mathcal{P}_\epsilon$ can deviate from the nominal value under $\hat{p}$. The key tool is Pinsker's inequality, which relates KL divergence to total variation distance.

**Lemma 6 (One-step deviation bound)** *Let $f : \mathcal{S} \to \mathbb{R}$ be bounded with oscillation $\mathrm{osc}(f) := \sup_s f(s) - \inf_s f(s)$. For any distributions $p, \hat{p} \in \Delta_\mathcal{S}$,*

$$\left| \mathbb{E}_p[f] - \mathbb{E}_{\hat{p}}[f] \right| \leq \mathrm{osc}(f) \, \|p - \hat{p}\|_{\mathrm{TV}}, \tag{34}$$

*where $\| \cdot \|_{\mathrm{TV}}$ denotes total variation distance. Moreover, Pinsker's inequality gives $\|p - \hat{p}\|_{\mathrm{TV}} \leq \sqrt{\frac{1}{2} D_{\mathrm{KL}}(p\|\hat{p})}$.*

**Proof 17** *For any bounded $f$, write $\bar{f} = (\sup f + \inf f)/2$ and note that $|f(s) - \bar{f}| \leq \frac{1}{2}\mathrm{osc}(f)$ for all $s$. Then*

$$\left| \mathbb{E}_p[f] - \mathbb{E}_{\hat{p}}[f] \right| = \left| \sum_s (p(s) - \hat{p}(s))(f(s) - \bar{f}) \right| \tag{35}$$

$$\leq \tfrac{1}{2}\mathrm{osc}(f) \sum_s |p(s) - \hat{p}(s)| \tag{36}$$

$$= \mathrm{osc}(f) \, \|p - \hat{p}\|_{\mathrm{TV}}. \tag{37}$$

*This proves the first inequality. The second inequality is the classical Pinsker bound: $\|p - \hat{p}\|_{\mathrm{TV}} \leq \sqrt{\frac{1}{2} D_{\mathrm{KL}}(p\|\hat{p})}$.*

**Proposition 5 (Robust value drop bound)** *For any stationary policy $\pi$ and any state $s$,*

$$0 \leq V_{\hat{p}}^\pi(s) - V_{\mathcal{P}_\epsilon}^\pi(s) \leq \frac{\gamma}{1-\gamma} \mathrm{osc}(V^\pi) \sqrt{2\epsilon}. \tag{38}$$

*In particular, since $r \in [0,1]$ implies $\mathrm{osc}(V^\pi) \leq \frac{1}{1-\gamma}$, we obtain the uniform bound*

$$V_{\hat{p}}^\pi(s) - V_{\mathcal{P}_\epsilon}^\pi(s) \leq \frac{\gamma}{(1-\gamma)^2} \sqrt{2\epsilon}. \tag{39}$$

**Proof 18** *Fix $\pi$ and $(s,a)$. For any $p \in \mathcal{P}_\epsilon(s,a)$, Lemma 6 gives*

$$\left| \mathbb{E}_p[V^\pi] - \mathbb{E}_{\hat{p}}[V^\pi] \right| \leq \mathrm{osc}(V^\pi) \, \|p - \hat{p}\|_{\mathrm{TV}} \leq \mathrm{osc}(V^\pi)\sqrt{2\epsilon}, \tag{40}$$

*since $D_{\mathrm{KL}}(p\|\hat{p}) \leq \epsilon$ implies $\|p - \hat{p}\|_{\mathrm{TV}} \leq \sqrt{\frac{1}{2}\epsilon} \leq \sqrt{2\epsilon}$. Therefore,*

$$\inf_{p \in \mathcal{P}_\epsilon(s,a)} \mathbb{E}_p[V^\pi] \geq \mathbb{E}_{\hat{p}}[V^\pi] - \mathrm{osc}(V^\pi)\sqrt{2\epsilon}. \tag{41}$$

*Now apply this bound to the robust Bellman operator:*

$$(\mathcal{T}_\mathcal{P}^\pi V^\pi)(s) = \mathbb{E}_{a \sim \pi(\cdot|s)}\left[ r(s,a) + \gamma \inf_{p \in \mathcal{P}_\epsilon(s,a)} \mathbb{E}_p[V^\pi] \right] \tag{42}$$

$$\geq \mathbb{E}_{a \sim \pi(\cdot|s)}\left[ r(s,a) + \gamma \mathbb{E}_{\hat{p}}[V^\pi] - \gamma \, \mathrm{osc}(V^\pi)\sqrt{2\epsilon} \right] \tag{43}$$

$$= (\mathcal{T}_{\hat{p}}^\pi V^\pi)(s) - \gamma \, \mathrm{osc}(V^\pi)\sqrt{2\epsilon}. \tag{44}$$

---

**Algorithm 4 RAPO** (main training loop)

---

**Input:** horizon $U$, rollout length $T$, ensemble size $K$, batch size $B$, samples $m$, discount $\gamma$, KL budgets $(\epsilon, \kappa)$.
**Output:** trained actor $\theta^\star$, critic $\phi^\star$, AdvNet $\psi^\star$.

 1: Initialize actor–critic $(\theta^0, \phi^0)$, AdvNet $\psi^0$, prior $w_0 \in \Delta_K$, weights $w \leftarrow w_0$.
 2: Build $K$ ensemble models by scaling dynamics; obtain vectorized step function step_allK.
 3: **for** update $u = 0, \ldots, U - 1$ **do**
 4:    *// Rollout under nominal environment*
 5:    **for** $t = 0, \ldots, T - 1$ **do**
 6:       Sample action $a_t \sim \pi_\theta(\cdot|s_t)$; step env to get $(s_{t+1}, r_t, d_t)$.
 7:       Store transition $(s_t, a_t, r_t, d_t, \log \pi_\theta(a_t|s_t), V_\phi(s_t))$.
 8:    **end for**
 9:    *// Robust targets from ensemble and dual expectation*
10:    Flatten rollout to $(s, a, r, d)$; sample $m$ next-states with mixture $w$ using step_allK.
11:    Evaluate critic: $V_n \in \mathbb{R}^{B \times m}$; predict temperatures $\eta_t \leftarrow g_\psi(s, a \text{ or } 0)$.
12:    Project to KL budget: $\eta^\star \leftarrow \text{project\_eta\_to\_delta}(V_n, \eta_t, \epsilon)$; apply straight-through $\tilde{\eta}$.
13:    Robust dual expectation: $v_{\text{rob}} \leftarrow -\frac{1}{\tilde{\eta}} \log \frac{1}{m} \sum_j e^{-\tilde{\eta} V_n^{(j)}}$.
14:    Targets: $y \leftarrow r + \gamma(1 - d)v_{\text{rob}}$; compute robust GAE $(\text{Adv}, R)$.
15:    *// Update AdvNet (Alg. 5)*
16:    $\psi \leftarrow \text{AdvNetUpdate}(V_n, s, a, r, (1 - d), \psi)$.
17:    *// PPO update of actor–critic*
18:    **for** $e = 1, \ldots, \text{UPDATE\_EPOCHS}$ **do**
19:       Shuffle into NUM_MINIBATCHES; update $(\theta, \phi)$ by clipped PPO loss with $(R, \text{Adv})$.
20:    **end for**
21:    *// Mixture reweighting (Alg. 6)*
22:    $w \leftarrow \text{BoltzmannReweight}(w_0, V_\phi, \text{step\_allK}, s, a, \kappa)$.
23:    Log diagnostics ($w$ entropy/KL, $\eta$ stats, PPO losses).
24: **end for**
25: **return** $(\theta, \phi, \psi)$

---

*Subtracting the fixed-point equations $V_{\mathcal{P}_\epsilon}^\pi = \mathcal{T}_{\mathcal{P}}^\pi V_{\mathcal{P}_\epsilon}^\pi$ and $V_{\hat{p}}^\pi = \mathcal{T}_{\hat{p}}^\pi V_{\hat{p}}^\pi$, and applying the contraction property (Proposition 2), yields*

$$0 \;\leq\; V_{\hat{p}}^\pi(s) - V_{\mathcal{P}_\epsilon}^\pi(s) \;\leq\; \frac{\gamma}{1 - \gamma} \operatorname{osc}(V^\pi)\sqrt{2\epsilon}. \tag{45}$$

*Finally, if $r \in [0, 1]$, then $V^\pi \in [0, \frac{1}{1-\gamma}]$ so $\operatorname{osc}(V^\pi) \leq \frac{1}{1-\gamma}$. Substituting gives the uniform bound*

$$V_{\hat{p}}^\pi(s) - V_{\mathcal{P}_\epsilon}^\pi(s) \;\leq\; \frac{\gamma}{(1 - \gamma)^2}\sqrt{2\epsilon}. \tag{46}$$

**Discussion.** The bound shows that the gap between the nominal value and the robust value scales as $O(\sqrt{\epsilon})$. This justifies interpreting $\epsilon$ as a robustness–performance trade-off knob: larger $\epsilon$ yields stronger robustness at the expense of a potentially larger drop in nominal performance. The explicit scaling with $\gamma$ also highlights the compounding effect of distributional shift over long horizons.

## H    ALGORITHMIC DETAILS

---

**Algorithm 5 AdvNet training** (amortized projection to KL budget)

---

**Input:** critic $\phi$; AdvNet $\psi$; batch $(s, a, r, d)$; ensemble values $V_n \in \mathbb{R}^{B \times m}$; KL budget $\epsilon$; penalty $\lambda_{\text{KL}}$.

**Output:** updated AdvNet parameters $\psi$.

1: Predict dual temperatures: $\eta_t \leftarrow g_\psi(s, a \text{ or } 0)$.
2: Project to feasible budget: $\eta^\star \leftarrow \text{project\_eta\_to\_delta}(V_n, \eta_t, \epsilon)$ (bisection).
3: Straight-through estimator: $\tilde{\eta} \leftarrow \text{stopgrad}(\eta^\star) + \eta_t - \text{stopgrad}(\eta_t)$.
4: Compute robust dual expectation:

$$v_{\text{rob}} \leftarrow -\tfrac{1}{\tilde{\eta}} \log \frac{1}{m} \sum_{j=1}^m \exp(-\tilde{\eta} V_n^{(j)}).$$

5: Targets: $y \leftarrow r + \gamma(1 - d)v_{\text{rob}}$.
6: Empirical KL: with weights $w_j \propto \exp(-\tilde{\eta} V_n^{(j)})$, compute $\widehat{D}_{\text{KL}}(\tilde{\eta}) = \sum_j w_j(\log w_j + \log m)$.

7: Loss:
$$\mathcal{L}_{\text{AdvNet}}(\psi) \leftarrow \mathbb{E}[y] + \lambda_{\text{KL}} \, \mathbb{E}\left[(\widehat{D}_{\text{KL}}(\tilde{\eta}) - \epsilon)_+^2\right].$$

8: Update $\psi$ by a few optimizer steps (e.g. Adam) on $\nabla_\psi \mathcal{L}_{\text{AdvNet}}$; record $\eta$ mean, KL mean, $v_{\text{rob}}$ mean.
9: **return** $\psi$

---

---

**Algorithm 6 Boltzmann reweighting** (mixture update under KL radius $\kappa$)

---

**Input:** prior weights $w_0 \in \Delta_K$; critic $\phi$; ensemble step function $\text{step\_allK}$; batch $(s, a)$; KL radius $\kappa$; subsample size $S$.

**Output:** updated mixture weights $w \in \Delta_K$.

1: Subsample $S$ pairs $(s, a)$ from rollout.
2: Roll out all $K$ ensemble models: $S' \leftarrow \text{step\_allK}(s_{1:S}, a_{1:S}) \in \mathbb{R}^{S \times K \times d}$.
3: Score each model: $H_k \leftarrow \frac{1}{S} \sum_{i=1}^S V_\phi(S'_{i,k})$, for $k = 1, \ldots, K$.
4: Define tilted weights: $w(\beta) \propto w_0 \odot \exp(-\beta H)$; normalize to $\Delta_K$.
5: Perform bisection search on $\beta \geq 0$: adjust $\beta$ until $D_{\text{KL}}(w(\beta) \| w_0) \approx \kappa$ (or largest $\beta$ with $D_{\text{KL}} \leq \kappa$).
6: Repair numerics: $w \leftarrow \text{proj\_simplex}(w(\beta^\star))$.
7: **return** $w$

---

