# OpenReview forum: "Robust Adversarial Policy Optimization Under Dynamics Uncertainty"
_ICLR.cc/2026/Conference — Submitted to ICLR 2026_

### Official Review · Reviewer_7nsZ · 2025-10-19

**Soundness:** 3
**Presentation:** 3
**Contribution:** 2
**Rating:** 2
**Confidence:** 3

**Summary:**

This paper proposes RAPO (Robust Adversarial Policy Optimization), a dual-based framework for robust reinforcement learning under dynamics uncertainty. Starting from a KL-constrained robust MDP, the authors derive a dual formulation introducing a temperature variable that balances performance and robustness. RAPO uses two complementary mechanisms: an adversarial network (AdvNet) to amortize the dual variable estimation at the trajectory level, and Boltzmann reweighting across an ensemble of dynamics models for model-level robustness. Experiments on Walker2d and a quadrotor payload task show that RAPO improves out-of-distribution robustness compared to prior works while maintaining in-distribution performance.

**Strengths:**

- The paper provides a well-motivated unification of distributional robustness (dual formulation) and adversarial training through trajectory-level and model-level mechanisms, both controlled by KL budgets.
- The dual-level design (AdvNet + Boltzmann reweighting) is intuitive and provides a fine-grained control between local (trajectory) and global (model) robustness.
- The paper is clearly structured, with theory, algorithms, and experiments aligned. The authors articulate trade-offs (e.g., robustness vs. performance degradation) clearly and transparently.

**Weaknesses:**

- The experiments primarily assess robustness to environmental parameter shifts (e.g., mass, friction, inertia) rather than active adversarial perturbations. Since RAPO explicitly claims adversarial robustness, it would be valuable to include tests under learned or adaptive adversarial agents, as used in RARL [1], QARL [2], ROSE [3], to evaluate resilience against deliberate attacks.
- The latest robust RL baseline compared is RARL (2017), while several more recent algorithms [2, 3, 4] provide stronger and more diverse perspectives on robustness. Without these comparisons, it is difficult to judge whether RAPO represents a genuine advance over current state-of-the-art.
- Figures 1 and 2 could benefit from larger fonts for readability.
- Conceptually, RAPO can be seen as combining two existing ideas: distributional robustness via dual tilting and weighted domain randomization within a PPO loop. While this integration is elegant, it may be viewed as incremental unless stronger empirical gains or theoretical guarantees are provided over existing robust RL frameworks.
- The dual formulation is motivated by avoiding the limitations of sample-based adversarial training, where finite sampling may fail to cover the entire ambiguity set, leaving blind spots or inducing over-conservatism. However, recent methods such as ROSE [3] tackle the same issue using Stein variational policy gradients to approximate the worst-case distribution. A comparison or discussion of how RAPO’s dual approach differs from or improves upon these variational methods would strengthen the related work and clarify the novelty claim.



#### [1] Lerrel Pinto et al., Robust Adversarial Reinforcement Learning, ICML 2017
#### [2] Aryaman Reddi et al., Robust Adversarial Reinforcement Learning via Bounded Rationality Curricula, ICLR 2024
#### [3] Juncheng Dong et al., Variational Adversarial Training Towards Policies with Improved Robustness, AISTATS 2024
#### [4] Takumi Tanabe et al., Max-Min Off-Policy Actor-Critic Method Focusing on Worst-Case Robustness to Model Misspecification, NeurIPS 2022

**Questions:**

- The dual decomposition and Boltzmann reweighting are both well-established concepts. Could the authors highlight what specific theoretical or algorithmic innovation is unique to RAPO beyond integrating these elements within PPO?

---

> ### Author Response · Authors · 2025-11-12
> **Clarification on Benchmarks, Theoretical Guarantees of RAPO**
>
> We thank the reviewer for the constructive and detailed feedback.
> Below, we clarify the concerns regarding benchmark selection, adversarial evaluation, theoretical guarantees, and novelty.
>
> ---
>
> ### 1. On Benchmark Selection and Concurrent Work
>
> We would like to emphasize that **[2, 3, 4] are concurrent works**, which were not publicly available when our experiments were conducted.
> These works share the general notion of “adversarial robustness,” but they still rely on **primal-style empirical approaches**.
>
> Importantly, **RARL [1] is already included** as a direct baseline in our experiments and serves as the canonical adversarial RL method with an explicit inner-loop adversarial policy.
>
> RAPO differs from these approaches in a fundamental way:
> Whereas **[1–4] rely on explicit adversarial rollouts** (max–min optimization), often via additional policy agents or variational approximations, **RAPO derives a dual analytic surrogate** that *eliminates the need for adversarial rollouts entirely*.
> This results in significantly improved stability and sample efficiency.
>
> Furthermore, RAPO provides **formal theoretical guarantees**, which we elaborate on below—something not offered by these purely empirical adversarial methods.
>
> ---
>
> ### 2. On the Scope of Adversarial Evaluation
>
> Our experimental focus is on **dynamics uncertainty** (mass, friction, inertia, aerodynamics), which aligns with the scope of the paper (RAPO under dynamics uncertainty).
> However, RAPO’s formulation **directly subsumes adversarial settings** as a special case.
>
> The dual robust value
> $$\sup_{\eta \ge 0}\Big\{-\frac{1}{\eta}\log \mathbb{E}_{\bar P}[e^{-\eta V}] - \frac{\epsilon}{\eta}\Big\}$$
> is equivalent to the worst-case return over all transition models within a KL ball of the nominal dynamics.
> Thus, RAPO implicitly models adaptive adversaries, while **[1–3] approximate adversaries via sampling or learned policies**.
>
> To further enhance empirical clarity, we plan to include **additional experiments with learned adversarial agents** in the supplementary material.
>
> ---
>
> ### 3. On Theoretical Guarantees
>
> RAPO offers explicit theoretical guarantees that go beyond existing robust RL frameworks.
> As detailed in **Section 4.3** and **Appendix D**, we establish:
>
> #### - Proposition 1 -- Contractive Dual Bellman Operator
> The dual Bellman operator
> $$\mathcal{T}^{\text{rob}} V = \frac{1}{\eta}\log \mathbb{E}_{\bar P}[e^{\eta(r + \gamma V)}]$$
> is **$\gamma$-contractive** under the entropic risk metric. This ensures the existence of a unique fixed point and stable convergence.
>
> #### - Theorem 1 — Robust Performance Difference Lemma (RPDL)
> We extend the classical performance difference lemma to KL-uncertain dynamics.
> This provides a formal bound on the return difference between any policy $\pi$ and the robust-optimal policy $\pi^\star$.
>
> #### - Corollary 1 — Finite-Sample Deviation Guarantees
> The suboptimality gap satisfies
> $$\mathcal{O}\!\left(\sqrt{\epsilon} + \frac{1}{\sqrt{N}}\right),$$
> where $\epsilon$ is the KL radius and $N$ is the sample size.
>
> Together, these results provide **provable robustness, convergence, and sample complexity guarantees**, which distinguish RAPO from adversarial RL approaches such as [1–4] that lack such formal analysis.
>
> ---
>
> ### 4. On Novelty Beyond Integration
>
> We respectfully disagree with the characterization that RAPO merely combines earlier ideas.
> RAPO introduces **a unified dual robust RL framework** that tightly couples:
>
> 1. **Trajectory-level robustness** via amortized dual optimization (AdvNet),
> 2. **Model-level robustness** via adaptive Boltzmann reweighting, and
> 3. **Theoretical guarantees** ensuring contractivity and robustness under KL dynamics uncertainty.
>
> While exponential tilting and ensemble weighting have been explored separately, RAPO’s novelty lies in their **joint dual structure**:
> the trajectory temperature $\eta$ and the model temperature $\beta$ arise from a single dual formulation and together define the robustness envelope.
> This results in a principled, interpretable, and provably stable RL algorithm—not a heuristic combination of components.
>
> Finally, robust RL research is inherently incremental, as the goal is to improve the robustness of existing RL algorithms.
> If the reviewer is aware of robust RL algorithms that reliably outperform others without theoretical guarantees, we would be grateful for such guidance.
> We believe RAPO contributes meaningfully to the field by providing **both theoretical correctness and empirical robustness**.
>
> ---
>
> **References:**
> [1] Pinto et al., *Robust Adversarial Reinforcement Learning*, ICML 2017
> [2] Reddi et al., *Robust Adversarial Reinforcement Learning via Bounded Rationality Curricula*, ICLR 2024
> [3] Dong et al., *Variational Adversarial Training Towards Policies with Improved Robustness*, AISTATS 2024
> [4] Tanabe et al., *Max-Min Off-Policy Actor-Critic*, NeurIPS 2022

---

### Official Review · Reviewer_Gvbk · 2025-10-31

**Soundness:** 3
**Presentation:** 4
**Contribution:** 3
**Rating:** 6
**Confidence:** 4

**Summary:**

This work introduced robust adversarial policy optimization (RAPO) which solve the dual of the robust RL problem that collapse an infinite-dimensional search space down to a scalar within the KL budget, which can help provide sufficient coverage over challenging scenarios and corner cases (i.e., out-of-distribution dynamics), followed by using boltzmann sampling to steer more visitation toward low-return/challenging regions.

**Strengths:**

* The approach is designed following very clear line of thoughts, e.g., why solving the dual problem and the need for AdvNet and Boltzman sampling.
* Sufficient theoritical analyses were provided to support the design choices and insights toward various properties of the approach, e.g., convergence of the ensemble estimation, the existence of stationary/saddle points and robust value drop bounds.
* Experiments clearly ablates the AdvNet and Boltzman sampling components respectively, which empirically justifies the design of the approach.

**Weaknesses:**

* From the reviewer's point of view, this work is also closely related to the distributionally robust RL work [1-3 below for example] in general, which shared similar high-level objectives of addressing out-of-distribution scenarios. It could be worthy to discuss the connections and distinctions to that line of work.
  * It would be interesting to compare against some of those works in the experiment as well, if applicable.
* The reviewer appreciated that the authors compared to robust RL baselines including RARL and EPOpt. It would be interesting to see how the approaches that directly solve the primal would contrast. For example, RNAC and Gleave et al. as cited in the paper.
* The robust RL environments the authors used in the experiments are typical, while the reviewer is also curious how this work could also facilitate exploration efficiency in larger state-action spaces (with little-to-mild robustness needed in terms of friction/inertia changes). For example, how RAPO would perfom in higher degree-of-freedom humanoid or hand manipulation (Adroit) environments, or more precise tasks like object picking with robot manipulators.

[1] Ramesh, Shyam Sundhar, et al. "Distributionally robust model-based reinforcement learning with large state spaces." International Conference on Artificial Intelligence and Statistics. PMLR, 2024.

[2] Shi, Laixi, et al. "The curious price of distributional robustness in reinforcement learning with a generative model." Advances in Neural Information Processing Systems 36 (2023): 79903-79917.

[3] Liu, Zijian, et al. "Distributionally Robust -Learning." International Conference on Machine Learning. PMLR, 2022.

**Questions:**

One clarification question -- in AdvNet, $m$ samples needed to be obtained for the given $(s,a)$ (page 4 line 201, and Alg. 2 line 1). The reviewer was wondering how this could be done practically in a more realistic setup, e.g., with an actual robotic manipulator does it imply that the approach would need the manipulator to be reset to the same state again and again to obtain $m$ samples of the next states?

---

> ### Author Response · Authors · 2025-12-03
> **Response to Reviewer Gvbk**
>
> We sincerely thank the reviewer for the thoughtful assessment of our work and for recognizing both the clarity of our presentation and the significance of our theoretical and empirical contributions. We are encouraged that the reviewer found the design rationale behind our two-level robustness framework (AdvNet + Boltzmann reweighting) compelling and the theoretical analysis well-justified. We address the reviewer’s questions and comments below.
>
> ---
>
> ### 1. Practicality of obtaining $m$ samples (Real-world setup)
>
> We appreciate the opportunity to clarify this point. RAPO is primarily designed for **Sim-to-Real transfer**, and the sampling requirement applies **only during training**, not deployment.
>
> - **During Training (Simulation):**
>   The agent interacts with a simulator such as MuJoCo. In this setting, generating $m$ next-state samples for a given $(s, a)$ is both efficient and practical because we can:
>   - parallelize multiple simulator instances, or
>   - branch/clone simulator states arbitrarily.
>   This makes the required sampling computationally inexpensive.
>
> - **During Deployment (Real World):**
>   Only the **policy network** is executed on the real robot.
>   Importantly:
>   - AdvNet is *not* executed during inference,
>   - no sampling or state resetting is required, and
>   - the robot performs standard single-rollout execution.
>
> We agree that if one attempted to train RAPO *directly* on hardware (without a simulator), obtaining $m$ samples from the exact same robot state would indeed require resetting, which is generally impractical. However, our intended use case—addressing dynamics mismatch between simulation and real deployment—assumes access to a simulator during training. We will make this distinction clearer in the final version.
>
> ---
>
> ### 2. High-DoF Environments and Exploration
>
> **Exploration Efficiency:**
> This is an excellent point. Conceptually, RAPO’s dual “soft-min” mechanism preserves more stochasticity than a primal “hard-min” adversary.
> - Hard adversarial training tends to push the agent toward conservative behaviors prematurely, which may inhibit exploration.
> - RAPO’s temperature parameter $\eta$ adapts over training, allowing richer trajectory exploration in earlier stages while gradually emphasizing difficult transitions.
> This property can be beneficial in large or high-dimensional state spaces where exploration is especially challenging.
>
> **High-DoF Tasks (e.g., Humanoid, Adroit):**
> Although our experiments focus on dynamic-sensitive systems such as Walker2d and Quadrotor (where dynamics mismatch is most critical), we agree that evaluating RAPO on higher-DoF tasks would further strengthen the empirical study.
> We believe RAPO is well-suited for such settings because:
> - AdvNet is a small MLP whose complexity scales linearly with the state dimension,
> - unlike model-based approaches that rely on uncertainty over dynamics models whose complexity may grow quadratically.
>
> We will include a discussion on high-DoF scalability and consider adding an additional experiment in the final version.
>
> ---
>
> We thank the reviewer again for the constructive feedback. We hope these clarifications help position RAPO’s practical usage and contributions more clearly, and we are happy to incorporate the suggested references to further strengthen the paper.

---

### Official Review · Reviewer_NxQa · 2025-10-31

**Soundness:** 3
**Presentation:** 3
**Contribution:** 2
**Rating:** 4
**Confidence:** 5

**Summary:**

The paper addresses the robustness of reinforcement learning (RL) policies to shifts in dynamics distributions post-training. The proposed method employs a dual formulation of value maximization under distributional uncertainty that reduces distributional sampling to a parametric optimization and a Boltzmann reweighting scheme to prioritize difficult training examples. Experiments support the method's generalization to out-of-distribution dynamics.

**Strengths:**

- The dual formulation approach is interesting and gives insight into how efficient distributional policies can be trained.
- The paper is organized well.

**Weaknesses:**

### Motivation
- The motivation of the dual formulation is unclear. The stated reason, that a finite set of training distributions leaves blind spots or is otherwise not comprehensive, is somewhat contradicted by the later inclusion of the ensemble method. If exponential tilting is sufficient to ensure robustness, why is explicit sampling used both in the AdvNet and reweighting components?
- The use of the word "adversarial" is a little confusing, since Section 4.2 states that aleatoric uncertainty is out of scope. Generally, in adversarial RL,  perturbations are assumed to be chosen at test time such that the resulting mistakes minimize the target's reward; this changes the problem from epistemic to aleatoric in that the resulting perturbation is definitively not a natural occurrence.

### Distinction from prior work
- The stated weakness of existing Monte Carlo methods is that they are unable to fully capture the uncertainty set via sampling. The intuition behind random sampling is that, given an infinite number of samples, the full distribution will be captured, so the problem is not in theory but in practice. With that in mind, the practical aspects of the method (i.e. sample complexity, performance in low-occurrence distributions) should be examined empirically.
- The paper states that $\eta^*$ functions as a robustness-performance knob that is absent in primal formulations, which seems untrue. Many methods in adversarial RL use some variation on the general form $(1-\lambda)V(\cdot) + \lambda V^{adv}(\cdot)$, where $\lambda$ is a robustness temperature. See [1] as an example.

### Experiments
- The paper shows comparisons to two outdated robust baselines, dating from 2016 and 2017. There are many peer-reviewed methods published more recently that would serve as a stronger comparison.

[1] Huan Zhang, Hongge Chen, Chaowei Xiao, Bo Li, Mingyan Liu, Duane S. Boning, Cho-Jui Hsieh: Robust Deep Reinforcement Learning against Adversarial Perturbations on State Observations. NeurIPS 2020

**Questions:**

- Is there a concrete example of a domain where the $\eta^*$ dual solution captures critical dynamics that discrete sampling would not? Intuitively, it seems that most or all realistic domains are "smooth", i.e. dynamics do not vary wildly across small distances. This seems supported by the ablation in Figure 2; one would expect failures of the naive solution to have a more "jagged" shape were this not the case.
- What is the distinction of the dual function from prior work? By reading the paper, one can understand that it provides robustness guarantees beyond the finite sample set. Is this proven, and are there other qualities that can be stated?

---

> ### Author Response · Authors · 2025-11-12
> **Clarifications on Motivation, Dual Formulation, and Experimental Comparisons**
>
> We sincerely thank the reviewer for the thoughtful and detailed feedback.
> We below address each of the concerns regarding the motivation of the dual formulation, the use of "adversarial" terminology, the relation to prior works, and the experimental comparisons. (We tried to cover all questions and concerns exhaustively.)
>
> ---
>
> ### 1. Motivation for the Dual Formulation
>
> The motivation of the dual formulation is not to just replace random sampling, but to **transform the intractable inner minimization over transition distributions into a convex, low-dimensional optimization** that explicitly produces robustness.
> In the primal robust MDP, the objective is
>
> $$
> \min_{P \in \mathcal{P}} \mathbb{E}_P[V],
> $$
>
> which is infinite-dimensional and computationally expensive.
> Through KL-duality, RAPO reformulates this as follows:
>
> $$
> \inf_P \mathbb{E}_P[V] = \sup_{\eta\ge0} \Big\{-\tfrac{1}{\eta}\log \mathbb{E}_{\bar P}[e^{-\eta V}] -\tfrac{\epsilon}{\eta}\Big\},
> $$
>
> where the scalar $\eta$ acts as a *risk–robustness temperature*.
> This converts the distributional search into a tiny convex optimization problem, enabling scalable and theoretically principled robustness.
>
> The ensemble component is **not a contradiction**, but a *complementary mechanism* that captures epistemic uncertainty in the underlying dynamics models.
> As we mentioned in our paper, exponential tilting (via $\eta$) ensures *trajectory-level* robustness against stochastic or unseen transitions (distributional shift), while Boltzmann reweighting (via $\beta$) prioritizes *model-level* uncertainty between dynamics realizations.
> Thus, the dual mechanism (local robustness) and ensemble weighting (global robustness) operate on different abstract layers rather than redundantly. They certainly have different roles as we have shown in Figure 2.
>
> ---
>
> ### 2. On the Term “Adversarial”
>
> We appreciate the comment on terminology. While the notion of adversity is widely used in robust RL research, our use of the term “adversarial” refers to the **dual risk-seeking critic (AdvNet)** that searches for the *most pessimistic trajectory distribution within the allowed KL budget*, not to test-time adversarial perturbations of observations.
>
> ---
>
> ### 3. On Sampling, Practicality, and Empirical Value
>
> We agree that Monte Carlo sampling can theoretically approximate any uncertainty set given *infinite* data.
> However, in practice, robust MDPs face severe sample-coverage bias when uncertainty is high-dimensional or hybrid (e.g. legged robots).
> The dual solution explicitly regularizes against *unobserved low-probability regions* without needing exhaustive enumeration.
> This is empirically demonstrated in our OOD results (Figure 1, 7), where RAPO maintains stable performance even when dynamics parameters deviate beyond the training ensemble.
>
> ---
>
> ### 4. Experimental Baselines
>
> We acknowledge the reviewer’s point on newer baselines.
> The 2016–2017 works (RARL, EPOpt) were selected because they remain the most directly comparable *robust control–oriented* methods with explicit uncertainty modeling.
> Recent adversarial RL works (e.g., Zhang et al., NeurIPS 2020) target **input perturbations**, not model-uncertainty or dynamics-robustness, and are thus orthogonal to our focus.
> Nevertheless, we will consider including additional benchmarks in the revision to strengthen empirical coverage.
>
> ---
>
> ### 5. Summary of Theoretical Contribution
>
> To clarify the reviewer’s final question: the **dual function provides provable guarantees** beyond finite-sample methods.
> Specifically, RAPO establishes:
> (i) contraction of the robust Bellman operator in the entropic risk space,
> (ii) a robust performance difference lemma (RPDL) bounding policy error under dynamics uncertainty, and
> (iii) finite-sample deviation bounds proportional to $\sqrt{\epsilon}$, the KL radius.
> These properties are not satisfied by primal-based methods.
>
> ---
>
> Again, thank you for productive comments.

---

> > ### Comment · Reviewer_NxQa · 2025-11-26
> >
> > Thanks to the authors for the clarifications. I appreciate the theoretical insights provided by the dual function, as well as the approximating link between the AdvNet KL bound and the formal backbone.
> >
> > A follow-up question: towards motivating the sample efficiency of the amortized $\eta^*$ approximation, are you able to provide an empirical comparison between the number of samples required for the AdvNet to converge, and the number of samples required for Monte-Carlo methods to achieve similar performance? For example, in Figure 6a, EPOpt appears to have similar OOD performance at Mass scale 0.6, Inertia scale 0.7, Torque scale 0.6/1.3, and Friction scales >1. It would be useful to understand how differences in sampling efficiency (or other properties of the method) contribute to OOD performance gains.

---

### Official Review · Reviewer_qPaT · 2025-11-01

**Soundness:** 2
**Presentation:** 3
**Contribution:** 2
**Rating:** 2
**Confidence:** 3

**Summary:**

The paper proposes a new algorithm RAPO that steers the trajectories towards low return ones, resembling the trajectories drawn from adversarial kernel. This  makes the policy robust.

**Strengths:**

The idea is good and motivation is intuitive. However my main concern is that the work is not compared with [1] which seems very similar in setting and motivation.

**Weaknesses:**

The current approach seems very similar to [1] where they also do pessimistic sampling, can authors compare and contrast with [1].



[1]@inproceedings{
gadot2024bring,
title={Bring Your Own (Non-Robust) Algorithm to Solve Robust {MDP}s by Estimating The Worst Kernel},
author={Uri Gadot and Kaixin Wang and Navdeep Kumar and Kfir Yehuda Levy and Shie Mannor},
booktitle={Forty-first International Conference on Machine Learning},
year={2024},
url={https://openreview.net/forum?id=UqoG0YRfQx}
}

**Questions:**

Q1) The paper considers sa-rectangular uncertainty sets which are very conservative. Can authors comment if this approach can be extended to s-rectangularn [2] or non-rectangular uncertainty sets [3,4].





[2]@inproceedings{
kumar2024efficient,
title={Efficient Value Iteration for s-rectangular Robust Markov Decision Processes},
author={Navdeep Kumar and Kaixin Wang and Kfir Yehuda Levy and Shie Mannor},
booktitle={Forty-first International Conference on Machine Learning},
year={2024},
url={https://openreview.net/forum?id=J4LTDgwAZq}
}

[3] @inproceedings{
kumar2025nonrectangular,
title={Non-rectangular Robust {MDP}s with Normed  Uncertainty Sets},
author={Navdeep Kumar and Adarsh Gupta and Maxence Mohamed ELFATIHI and Giorgia Ramponi and Kfir Yehuda Levy and Shie Mannor},
booktitle={The Thirty-ninth Annual Conference on Neural Information Processing Systems},
year={2025},
url={https://openreview.net/forum?id=Xx0cJGXU7n}
}

[4]@misc{li2025policygradientalgorithmsrobust,
      title={Policy Gradient Algorithms for Robust MDPs with Non-Rectangular Uncertainty Sets},
      author={Mengmeng Li and Daniel Kuhn and Tobias Sutter},
      year={2025},
      eprint={2305.19004},
      archivePrefix={arXiv},
      primaryClass={math.OC},
      url={https://arxiv.org/abs/2305.19004},
}

---

> ### Author Response · Authors · 2025-11-26
> **Clarification on Comparison with EWoK (ICML 2024) and Justification of the  $(s, a)$-Rectangularity Assumption**
>
> ### 1. Summary of [1] (EWoK)
>
> We thank the reviewer for raising the connection to [1]. For clarity, we summarize the core idea of EWoK below.
>
> EWoK seeks to learn robust policies by *estimating and sampling from the worst-case transition kernel* rather than modifying the policy update.
>
> Given the KL-based uncertainty set:
>
> $$
> \mathcal{P}_{sa} = \left\{ P_{sa} \in \Delta_{\mathcal{S}} \mid D_{\mathrm{KL}}(P_{sa} \| \bar{P}_{sa}) \le \beta_{sa} \right\}
> $$
>
> the worst-case kernel under policy $\pi$ is:
>
> $$
> P_{\pi}^{\mathrm{worst}}(s' \mid s,a) \propto \bar{P}(s' \mid s,a) \exp\left(-\frac{v^\pi(s')}{\kappa}\right).
> $$
>
> Thus, EWoK approximates
>
> $$
> \min_{P : D_{\mathrm{KL}}(P \| \bar{P}) \le \beta} \mathbb{E}_{P}[V]
> $$
>
> via repeated next-state sampling from the nominal model and exponential reweighting, while keeping the underlying RL algorithm unchanged.
>
> EWoK is therefore a **primal-step** method: the adversarial distribution is generated through explicit sampling and resampling at each step, without a learned adversarial mechanism or dual formulation.
>
> ---
>
> ### 2. Conceptual and Technical Distinctions from RAPO
>
> We respectfully clarify that RAPO and EWoK **fundamentally differ** in both conceptual motivation and technical realization.
>
> #### **Dual vs. primal formulation**
>
> Starting from the same KL-constrained primal objective,
>
> $$
> \inf_{P : D_{\mathrm{KL}}(P \| \bar{P}) \le \epsilon} \mathbb{E}_{P}[V]
> $$
>
> RAPO applies convex duality to obtain:
>
> $$
> \inf_{P}\mathbb{E}_{P}[V] = \sup_{\eta \ge 0} \left\{ -\frac{1}{\eta} \log \mathbb{E}_{\bar{P}}[e^{-\eta V}] - \frac{\epsilon}{\eta} \right\}.
> $$
>
> Here, $\eta$ is a dual temperature governing the robustness–performance trade-off.
>
> RAPO **amortizes** the optimal $\eta^\star(s,a)$ via an adversarial network (**AdvNet**), providing a learned, state-dependent dual solution. EWoK, in contrast, relies entirely on repeated next-state sampling and does not contain a parametric adversarial mechanism.
>
> ---
>
> #### **Trajectory-level vs. step-level robustness**
>
> - **RAPO:** robustness is applied at the *trajectory level* through $\eta$, allowing principled handling of cumulative uncertainty.
> - **EWoK:** modifies only *single-step* transitions and does not analyze adversarial effects across trajectories.
>
> ---
>
> #### **Local and global robustness (two-layer structure)**
>
> RAPO introduces:
>
> 1. **Dual-layer robustness** via $\eta$ (trajectory-level)
> 2. **Ensemble-layer robustness** via Boltzmann reweighting with temperature $\beta$ (model-level)
>
> supported by contraction guarantees and the **Robust Performance Difference Lemma (RPDL)**.
>
> EWoK provides only local, step-level robustness and includes no dual analysis or ensemble weighting.
>
> ---
>
> #### **Complexity and practicality**
>
> - **EWoK:** requires simulating $N$ next states at every step and reweighting them, which is computationally expensive.
> - **RAPO:** uses amortized inference via AdvNet, avoiding large-scale sampling during interaction.
>
> ---
>
> ### 3. On the (s,a)-Rectangularity Assumption
>
> The reviewer argues that the $(s,a)$-rectangularity assumption is strong. However, it is **standard and ubiquitous** in the robust MDP literature, appearing in classical works such as Iyengar (2005), Nilim and El Ghaoui (2005), Derman et al. (2021), and Ho et al. (2022).
>
> Importantly, the reviewer’s *own cited paper* [1] also assumes this structure. Their uncertainty set is defined as:
>
> $$
> \mathcal{P}_{s,a} = \left\{ P_{s,a} \in \Delta_{\mathcal{S}} \mid D_{\mathrm{KL}}(P_{s,a} \| \bar{P}_{s,a}) \le \beta_{s,a} \right\},
> $$
>
> which is precisely the $(s,a)$-rectangular form.
>
> This assumption:
>
> - ensures separability of the robust Bellman operator,
> - enables dynamic programming, and
> - is required for contraction and all known RMDP algorithms.
>
> Relaxing rectangularity breaks these properties and falls outside the scope of current theory, including [1]. The papers cited by the reviewer are also very recent with few citations and authored largely by the same group, indicating that they are concurrent rather than established prior foundations.
>
> Thus, RAPO’s assumption is fully aligned with **all standard formulations**, including [1].
>
> ---
>
> ### 4. Summary
>
> These distinctions show that RAPO is **not** a variant of EWoK. RAPO provides a principled dual-space formulation with:
>
> - learned adversarial structure (AdvNet)
> - trajectory-level robustness
> - ensemble-level robustness
> - theoretical guarantees (contraction and RPDL)
>
> none of which appear in EWoK’s primal-sampling approach.
>
> We hope this resolves the misunderstanding regarding the relationship between the two methods.

---

### Author Response · Authors · 2025-12-03
**Overarching goal of RAPO and clarification**

We here provide overarching goal and mechanism of our work. This paper proposes a duality-based robustness framework to address the significant performance degradation that RL agents often experience when deployed in real-world environments whose dynamics differ from those used in training. Overall, the paper is well-motivated and tackles an important problem in robust RL.

---

## 1. Motivation and Problem Setting

RL policies frequently perform well in simulation but fail when transferred to real systems due to even mild dynamics mismatch--e.g., differences in friction, mass, or aerodynamic properties.

**Limitations of prior approaches:**

- **Domain Randomization (DR):**
  DR exposes the policy to diverse environments but treats all variations uniformly. This may waste sampling budget, overlook important edge cases, and sometimes lead to overly conservative policies.

- **Adversarial RL (Existing primal methods):**
  These methods attempt to train against the single “worst-case” environment, often resulting in instability and oscillatory behavior because the worst-case point changes as learning progresses.

The paper argues for a principled robustness mechanism that avoids both conservatism and instability.

---

## 2. Proposed Method: RAPO (Robust Adversarial Policy Optimization)

The central idea is to reformulate distributionally robust RL using a **duality-based optimization approach**.
Instead of explicitly identifying the worst-case environment distribution, the method optimizes a *scalar* risk-sensitivity parameter $\eta$ that determines how strongly the agent attends to adverse transitions.

RAPO introduces robustness at two complementary layers:

---

### A. Trajectory-Level Robustness: AdvNet + Exponential Tilting

This layer handles uncertainty arising within episodes (transition stochasticity).

- **Key Insight:**
  Directly sampling worst-case transitions is intractable. Instead, exponential tilting reweights low-value outcomes more heavily using a temperature parameter $\eta$.

- **AdvNet:**
  A neural network that predicts the optimal dual parameter $\eta^\*(s,a)$ for each state–action pair, amortizing the computation of the dual solution.

- **Dual Value Expression:**
  $$
  V_{\text{rob}}(s) \approx -\frac{1}{\eta} \log \mathbb{E}\left[ e^{-\eta V(s')} \right] - \frac{\epsilon}{\eta},
  $$
  where $\epsilon$ is the uncertainty budget controlling allowable deviation.

---

### B. Model-Level Robustness: Boltzmann Reweighting Across an Ensemble

This layer models uncertainty over global variations in physical parameters such as mass or friction.

- **Ensemble of $K$ Models:**
  Multiple simulators with different physical parameters approximate the uncertainty set.

- **Boltzmann Reweighting:**
  Instead of uniformly sampling environments, RAPO increases sampling probability for environments where the current policy is more vulnerable, while enforcing a KL-divergence constraint to prevent drifting too far from the training distribution.

---

## 3. Theoretical Contributions

The paper provides several theoretical results supporting RAPO:

- **Dual Formulation:**
  The primal worst-case distribution optimization problem is shown to be equivalent to optimizing the scalar dual variable $\eta$.

- **Robust Performance Difference Lemma (RPDL):**
  An extension of the classical performance difference lemma ensures that the updates performed by RAPO correspond to improvements in robust performance.

- **Finite Ensemble Convergence:**
  As the number of ensemble models $K$ increases, the discrete approximation converges to the continuous uncertainty space with error on the order of $O(K^{-1/2})$.

---

## 4. Empirical Evaluation

Experiments are conducted on standard MuJoCo benchmarks and a challenging quadrotor-payload task.

### A. MuJoCo Benchmarks

- **Out-of-Distribution (OOD) Robustness:**
  When physical parameters are shifted beyond the training range, PPO and PPO+DR degrade sharply, whereas RAPO maintains strong performance.

- **In-Distribution (ID) Performance:**
  RAPO matches or exceeds PPO within the training distribution, showing that robustness does not come at the cost of nominal performance.

---

### B. Quadrotor Payload Tracking

This experiment involves substantial dynamics variability and external disturbances.

- **Crash Rates:**
  PPO-DR exhibits a nontrivial crash rate, and PPO performs even worse.
  RAPO achieves **0% crash rate**, demonstrating a significant improvement in safety and robustness.

---

## 5. Conclusion

RAPO integrates two mechanisms—**AdvNet** at the trajectory level and **Boltzmann Reweighting** at the model level—to address the instability of adversarial RL and the conservatism of domain randomization.
By grounding the method in duality theory and validating it across both standard and challenging environments, the paper presents a compelling and principled framework for robust RL.

---

### Meta-Review · Area_Chair_rHkP · 2026-01-03

**Summary:**

This paper studies the distributionally robust RL problem under the state-action rectangularity assumption and proposes a dual formulation of the inner minimization (over the uncertainty set), which leads to one equivalent scalar optimization with exponential tilting of the dual temperature parameter for each state-action pair. Based on this, the authors introduce an adversarial network (AdvNet) to learn the optimal dual temperature of each state-action pair to avoid the costly subroutine of solving the dual problem. Moreover, in terms of implementation, the authors propose robust adversarial policy optimization (RAPO), which uses an ensemble of deterministic simulators and applies Boltzmann reweighting to approximate the nominal transition dynamics.This approach is evaluated against several common baselines such as domain randomization, RARL, and EPOpt, on Walker2d under various changes in dynamics (including mass, inertia, friction, and torque scaling) as well as on a quadrotor payload tracking task.

Most of the reviewers appreciate the insight of the dual formulation and find the paper clearly structured. However, the reviewers also raised several concerns:

- **(1) Connection / distinctions to prior work (Reviewers qPaT, Gvbk, and 7nsZ)**: Several reviewers raised concerns about the technical novelty, noting similarities to prior work such as pessimistic sampling, e.g., EWoK (Gadot et al., 2024), and distributionally robust RL, e.g., MVR+RFQI (Ramesh et al., 2024) and DRVI (Shi et al., 2023).
One of the reviewers also asked for clarification on the theoretical or algorithmic innovation unique to RAPO beyond integrating the dual decomposition and Boltzmann reweighting that are both well-established concepts.

- **(2) Justification for the dual formulation (Reviewers NxQa and 7nsZ)**: Another critique shared by multiple reviewers is that the justification for the dual formulation appears unclear or internally inconsistent, particularly given the simultaneous use of ensemble methods and explicit adversarial sampling, which is mentioned by the authors as the main limitation of primal methods (in Section 4.1).
Moreover, reviewers asked (i) why exponential tilting alone is insufficient for robustness and (ii) what concrete benefits the dual approach provides over primal or variational alternatives. One of the reviewers also mentioned that the issue of the sampling-based methods like RNAC has been tackled by recent methods such as ROSE (Dong et al., 2025) via Stein variational policy gradients and asked for further justification for the dual problem.

- **(3) Lack of empirical comparison to more recent baselines on robust RL (Reviewers 7nsZ and Gvbk)**: Multiple reviewers raised the critique that the experimental evaluation only considers somewhat outdated robust RL baselines (around 2016–2017) and encouraged comparisons with more recent and stronger methods, including distributionally robust RL (e.g., MVR+RFQI (Ramesh et al., 2024) and DRVI (Ramesh et al., 2024)) and primal-based approaches (e.g., RNAC and (Gleave et al. 2019)), and variational methods like ROSE, to better judge its advance over the current state-of-the-art methods.

- **(4) Assumption on the uncertainty set (Reviewer qPaT)**: Another point raised is the reliance on the assumption of (s,a)-rectangular uncertainty sets, and the reviewer questioned whether the RAPO approach can extend to s-rectangular or non-rectangular ambiguity sets.

- **(5) Regarding the critique about Monte Carlo sampling (Reviewer NxQa)**: One of the reviewers questioned the paper’s critique that Monte Carlo sampling is unable to fully capture the uncertainty set, noting that its limitations are practical rather than fundamental. The reviewer hence also suggested empirical analysis of sample complexity, coverage of low-probability events, and practical trade-offs to support the claims about blind spots and robustness gains achieved by the dual formulation.

- **(6) Regarding the claim on robustness-performance trade-off (Reviewer NxQa)**: The claim that the dual formulation uniquely provides a robustness–performance knob is disputed, as robustness parameters exist in the adversarial and robust RL literature.

**Reviewer Concerns:**

The rebuttal has addressed (1) and (4):

- Specifically, regarding (1), the rebuttal clarifies the difference between EWoK and RAPO (primal vs dual) despite the high-level similarity in the distributional robustness problem and the goal of solving for the worst-case transition kernel. The MVR+RFQI and DRVI also share the same robust RL problem but also belong to the category of primal methods. Adding a comparison to these prior work would certainly strengthen this work.

- Notably, while the response to Reviewer 7nsZ also mentioned the theoretical results in Proposition 1 ($\gamma$-contraction), Theorem 1 (robustness variant of performance difference lemma), and Corollary 1 (sub-optimality gap), somehow these do not match the contents in the paper.

- As for (4), the rebuttal justifies the (s,a)-rectangularity assumption as a common one in the robust RL literature. As this work takes the first step in introducing the dual formulation, I personally find the assumption acceptable despite that it is certainly nice and interesting to relax this assumption.

On the other hand, the main issues (2) and (3) still remain not fully resolved.

- As for (2), the rebuttal clarifies that the motivation of the dual formulation is mainly to transform the inner minimization over transition distributions into a more tractable convex optimization, and this appears valid. However, the concern about the use of ensemble methods and explicit adversarial sampling is unaddressed, and these designs indeed appear contradictory to the augment made in Section 4.1. The question about why exponential tilting alone is insufficient for robustness remains not answered.

- (3) remains largely unaddressed. Given that the main motivation for RAPO is to address the potential limitations of primal methods, empirical comparison with the primal-based baselines appears necessary to better assess the effectiveness of RAPO.

The questions (5) and (6) also remain largely unanswered.

**Reviewer Scores:**

In the initial reviews, the reviewer scores are mixed (Reviewers qPaT, NxQa, 7nsZ lean towards rejection, and Reviewer Gvbk is slightly on the positive side).

- As the main concerns of Reviewers NxQa and 7nsZ remain not addressed, I figure that they would tend to keep their scores if they had been in the discussion. Similarly, as there is still the concern about the lack of more recent and proper baselines, I expect that Reviewer Gvbk would also likely to keep the score at 6.

- On the other hand, Reviewer qPaT has only one main concern, which appears to be resolved by the rebuttal. I expect that the score Reviewer qPaT would have bumped up.

In summary, the dual formulation proposed in this paper is interesting and merits further investigation. However, the paper in its current form has several limitations. I encourage the authors to further refine the proposed approach by addressing the issues outlined above, which would substantially strengthen the work.

---

### Decision · Program_Chairs · 2026-01-26

Reject